# Higgs Condensates are
# Symmetry-Protected Topological Phases:
## I. Discrete Symmetries

Ruben Verresen[1], Umberto Borla[2,3], Ashvin Vishwanath[1], Sergej Moroz[4,5] and Ryan Thorngren[6]

**1** Department of Physics, Harvard University, Cambridge, MA 02138, USA
**2** Physik-Department, Technische Universit ät München, 85748 Garching, Germany
**3** Munich Center for Quantum Science and Technology, 80799 München, Germany
**4** Department of Engineering and Physics, Karlstad University, Karlstad, Sweden
**5** Nordita, KTH Royal Institute of Technology and Stockholm University, Stockholm, Sweden
**6** Kavli Institute of Theoretical Physics, University of California, Santa Barbara, California 93106, USA

March 18, 2024

## Abstract

**Where in the landscape of many-body phases of matter do we place the Higgs condensate of a gauge theory? On the one hand, the Higgs phase is gapped, has no local order parameter, and for fundamental Higgs fields is adiabatically connected to the confined phase. On the other hand, Higgs phases such as superconductors display rich phenomenology. In this work, we propose a minimal description of the Higgs phase as a symmetry-protected topological (SPT) phase, utilizing conventional and higher-form symmetries. In this first part, we focus on 2+1D $\mathbb{Z}_2$ gauge theory and find that the Higgs phase is protected by a higher-form magnetic symmetry and a matter symmetry, whose meaning depends on the physical context. While this proposal captures known properties of Higgs phases, it also predicts that the Higgs phase of the Fradkin-Shenker model has SPT edge modes in the symmetric part of the phase diagram, which we confirm analytically. In addition, we argue that this SPT property is remarkably robust upon explicitly breaking the magnetic symmetry. Although the Higgs and confined phases are then connected without a bulk transition, they are separated by a boundary phase transition, which we confirm with tensor network simulations. More generally, the boundary anomaly of the Higgs SPT phase coincides with the emergent anomaly of symmetry-breaking phases, making precise the relation between Higgs phases and symmetry breaking. The SPT nature of the Higgs phase can also manifest in the bulk, e.g., at transitions between distinct Higgs condensates. Finally, we extract insights which are applicable to general SPT phases, such as a 'bulk-defect correspondence' generalizing discrete gauge group analogs of Superconductor-Insulator-Superconductor (SIS) junctions. The sequel to this work will generalize 'Higgs=SPT' to continuous symmetries, interpreting superconductivity as an SPT property.**

# 1  Introduction

The importance of the Anderson-Higgs [1–4] mechanism across different areas of physics can hardly be overstated. The phases arising from the condensation of charged matter fields in a gauge theory, simply referred to here as Higgs phases, can be found throughout physics, ranging from the electroweak sector of the Standard Model to the superconducting state of electronic materials.

**What is the Higgs phase?** Despite its importance and decades of theoretical studies, certain conceptual aspects of Higgs phases remain unclear. For instance, in the modern approach to quantum phases of matter, we employ symmetry principles, topology, and their interplay to classify quantum states separated by quantum phase transitions. In such an approach, symmetry breaking phases are classified by order parameters which transform under the symmetry; these are colloquially referred to as being "condensed" in the ordered phase, in analogy with Bose-Einstein condensation. While the Higgs phase superficially resembles such a condensate, the distinction between symmetry charge and gauge charge precludes a full analogy. For instance, it is well known that this distinction leads to the absence of gapless Goldstone modes in the latter case. This raises the question—*what is the broader category of quantum states within which we should include the Higgs phase?* A satisfactory answer is not just conceptually desirable but could also uncover new properties of the Higgs phase as well as provide a unifying framework for its known properties.

A potential answer to this question, at least for a Higgs phase obtained by condensing a *fundamental* representation of the gauge group (colloquially, matter with unit gauge charge), is that the Higgs phase is 'trivial'—a gapped phase that has no interesting properties. More precisely, in a gauge theory model embedded in a tensor product Hilbert space of sites, we may say that a 'trivial' state can be smoothly connected to a simple product state, without encountering any singularities along the way. Indeed, a seminal work by Fradkin and Shenker showed that a unit-charge Higgs phase of a lattice gauge theory is connected to the confined phase, and may as well be considered trivial; see Fig. 1(a) where we reproduced the particular case of a $\mathbb{Z}_2$ gauge group in 2+1D, although qualitatively the same phase diagram applies to $U(1)$ gauge theory in 3+1D [7]. This point of view is in tension with a different observation. It is well-known that superconductors exhibit a variety of remarkable phenomena ranging from dissipationless supercurrents to Josephson effects. Viewed as a Higgs condensate (for $U(1)$ gauge theory) this begs the question of how a trivial gapped phase is endowed with these remarkable properties?

**Two new ingredients: SPTs and higher symmetries.** Although these issues are not new, two recent conceptual developments will allow us to sharpen these questions and unify the above two seemingly-contradicting observations. First, there has been an enormous growth of our understanding of *symmetry-protected topological (SPT) phases*—ranging from the Haldane-Affleck-Kennedy-Lieb-Tasaki (Haldane-AKLT) spin-1 chain in 1+1D to topological insulators and superconductors as well as interacting SPTs in different dimensions [8–18]. A key takeaway from this body of work is that in a variety of

(a) periodic b.c.                                    (b) open b.c.

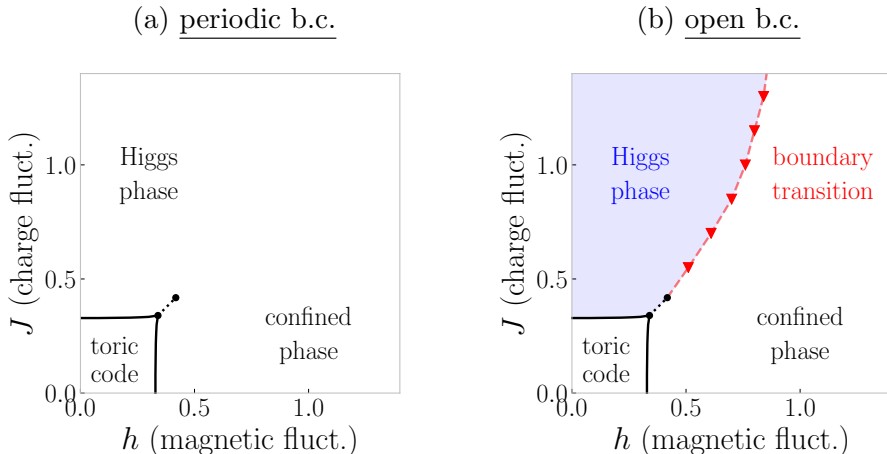

Figure 1: **Edge modes in the Higgs phase of $\mathbb{Z}_2$ gauge theory**. (a) The phase diagram of the Fradkin-Shenker model for $\mathbb{Z}_2$ lattice gauge theory in 2+1 dimensions (reproduced from Ref. [5]), where the Higgs condensate (for fundamental matter) and the confined phase are adiabatically connected. Strictly speaking, this is the phase diagram for periodic boundary conditions. We label the deconfined phase as the 'toric code' phase [6]. (b) Upon including rough boundaries (which respect the so-called magnetic symmetry; see Fig. 2), the Higgs and confined phase are separated by a boundary phase transition. The blue shaded region has degenerate edge modes (Sec. 4.4.3). This traces back to the fact that in the $h = 0$ limit, the Higgs phase is a non-trivial SPT phase protected by matter and magnetic symmetry. For other boundary choices, see Sec. 4.4.5. A similar phase diagram applies to $U(1)$ gauge theory in 3+1D.

circumstances, novel phases can be distinguished in ways that are fundamentally different from that of broken symmetries. For instance, a nonlocal string order parameter detects the Haldane-AKLT phase, and protected edge modes are a ubiquitous feature of SPTs, as long as symmetries are preserved.

Second, the notion of symmetries and symmetry breaking has been extended to include *higher(-form) symmetries* [19, 20]. These allow us to view intrinsic topological order—which is outside the paradigm of *conventional* symmetry breaking—as a formal extension of conventional (global or 0-form) symmetries, to higher $p$-form symmetries, where $0 < p \leq d$, the latter being the dimension of space. In this language, the deconfined phase of the $\mathbb{Z}_2$ gauge theory for $d \geq 2$, which is characterized by toric code topological order, spontaneously breaks a $(d-1)$-form symmetry. A remarkable property of such higher symmetries, which is not shared by conventional (0-form) symmetries, is that their physical consequences at low energies are generically *stable* to weak explicit breaking. In contrast, breaking of a regular symmetry typically removes its effect at the lowest energies. For example, Goldstone modes for 0-form symmetries acquire a gap when symmetries are explicitly broken, even if the breaking is weak. In contrast, the 'Goldstone modes' of the spontaneously broken *higher* symmetries in the deconfined phase of 3+1D $U(1)$ gauge theory do not require any fine-tuning at the microscopic scale; these are the massless photons of our universe.

The interplay between these two developments has thus far only been partially explored. In particular, it has been realized that a generalization of SPT phases that involves *both* global and higher-form symmetries is quite natural [21–26]. Again, such phases have neither symmetry breaking nor intrinsic topological order, but exhibit edge states. The options for edge states can differ depending on the higher symmetries involved. However, thus far such cases have only been explored with *exact* higher symmetries, in which case

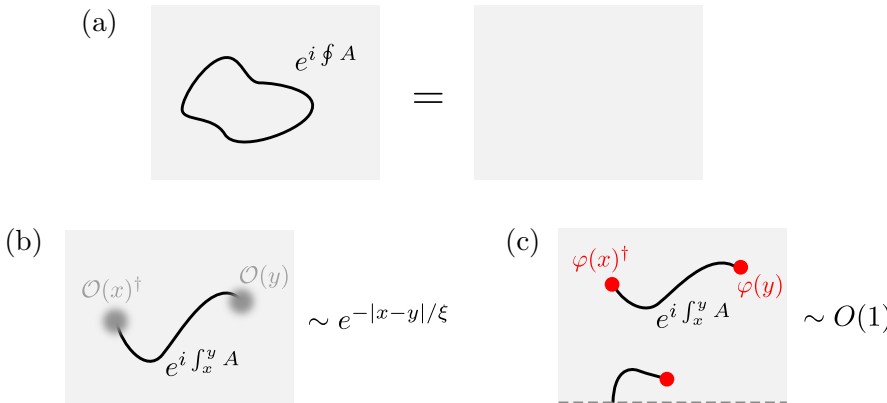

Figure 2: **Magnetic symmetry, higher-form symmetry breaking, and SPT string order for the Higgs phase.** (a) If magnetic flux is preserved, $\mathbb{Z}_n$ gauge theory has a magnetic symmetry generated by the Wilson line on closed loops, leaving the ground state invariant. (b) The deconfined phase spontaneously breaks this stringlike ('higher') symmetry. This means *open* Wilson strings having exponentially-decaying correlations for any endpoint operator $\mathcal{O}(x)$. Physically, disorder operators are *not* condensed. (c) In the Higgs phase, magnetic symmetry is restored. Due to gauge invariance, the Wilson string can only have long-range order if its endpoint carries a matter field. This endpoint is charged under the global matter symmetry, which we interpret as a non-trivial SPT invariant. If Wilson strings can end at the boundary (dashed line), this implies a ground state degeneracy since we have a nonzero v.e.v. for a charged operator. (While the above can apply to $U(1)$ gauge theory, it depends on which type of magnetic symmetry one enforces [27].)

they are of limited physical relevance. Curiously, it has not yet been explored whether the phenomenology of SPT phases protected by higher symmetries is robust to explicitly breaking the latter—perhaps because a natural physical situation for having such higher SPTs had not yet been recognized.

**Higgs=SPT.** In this work, we bring together the above developments and argue that *the Higgs phase should properly be viewed as a symmetry protected topological phase* (which we refer to as 'Higgs=SPT' in short), potentially involving higher symmetries as determined by the gauge group and spacetime dimensionality. We will see that this viewpoint succinctly captures known universal properties of Higgs phases arising from a variety of gauge groups, and also predicts new properties that are numerically verified in this work. Furthermore, by embedding Higgs phases within the broader and well-known family of quantum phases of matter both helps with a deeper understanding of the phase and its interplay with other states that can be coupled to it.

An important ingredient for Higgs=SPT will be the presence of "magnetic" symmetry, that in conjunction with a global symmetry related to the condensing Higgs field, leads to the nontrivial SPT phase. As a minimal example, one can consider $\mathbb{Z}_2$ gauge theory in $d \geq 1$ spatial dimensions, where the electric charges play the role of the Higgs field. In the absence of dynamical magnetic fluxes, it can be said to have a magnetic $(d-1)$-form symmetry. The symmetry generators are the Wilson loops $e^{i \oint_\gamma A}$ supported on closed curves $\gamma$ as shown in Fig. 2(a). For $U(1)$ Maxwell theory, the magnetic symmetry is simply the absence of magnetic monopoles, which for this continuous gauge group leads to a $(d-2)$-form symmetry.

The slogan Higgs=SPT captures two main claims (both of which imply a host of physical results). The first is that in the presence of the aforementioned magnetic symmetry, the Higgs phase is a non-trivial SPT phase. In particular, the symmetries give rise to a

bulk topological invariant, which in turn implies protected edge or interface modes, as we will discuss in detail. In fact, as we will discuss in Part II [27], supercurrents and the Josephson effect turn out to be direct consequences of this SPT perspective. The second claim pertains to the cases where the magnetic symmetry is a higher-form symmetry: upon explicitly breaking this symmetry (which is rather natural from the perspective of lattice gauge theory), the physical consequences of the SPT phase are parametrically robust— despite the phase itself strictly speaking now being trivial. The triviality statement agrees with the known fact that with periodic boundary conditions, the Fradkin-Shenker phase diagram adiabatically connects the Higgs and confined phases; e.g., see Fig. 1(a) for $\mathbb{Z}_2$ gauge theory in $d = 2$. However, the parametric stability implies the new result that in the presence of certain boundary conditions, the Higgs and confined phases are necessarily separated by a *boundary phase transition* (see Fig. 1(b)).

We demonstrate and confirm Higgs=SPT for a variety of gauge groups and dimensionalities, using a combination of analytic field-theoretic and lattice arguments, as well as numerical simulations. In particular, in the present volume we focus on the lattice gauge theories with discrete symmetries, whereas we discuss continuous gauge groups and superconductors in Part II [27]. Before delving into those case studies, we start with a bird's-eye view of Higgs=SPT in Sec. 2. There we also highlight a variety of subtleties, one of which is the fact that the SPT phase is protected in part by the matter symmetry, which might seem strange given that it is seemingly a gauge symmetry (and thus unphysical). Nevertheless, we discuss a host of situations where the matter symmetry acts on local physical degrees of freedom in the theory. We end that section with an outline of the remainder of this work (Sec. 2.2).

**Prior work.** Some of the essential ingredients for Higgs=SPT have been around a long time, but never completely assembled. At a high level, SPTs may be thought of as condensates of non-local charged objects such as domain walls [28] or vortices [29,30]. One aproach more obviously related to ours is described in Ref. [31], where the authors used anyon condensation to construct 0-form SPTs. They considered a deconfined discrete gauge theory with a fractionalized global symmetry $G$. Condensing anyons in such a theory tends to produce non-trivial $G$-SPT phases. They studied a confined phase, but one could just as easily condense charges in their picture. In a more recent work still closer in spirit to ours [32], the authors studied the same mechanism for certain crystalline symmetries, but emphasized the role played by the Gauss law as an SPT stabilizer when charged anyons in particular are condensed.

The aim of our work is different: here we do not seek to enrich a gauge theory with additional symmetries as means of generating new states of matter, but rather we consider the symmetries which are already naturally present in order to make claims which broadly apply to Higgs phases thereby uncovering their true nature. Moreover, doing so, we go full circle and find new insights which are more generally applicable to SPT phases.

This research program was initiated by some of the present authors in Ref. [33], which studied the one-dimensional case. There it was indeed found that the Higgs phase of $\mathbb{Z}_2$ gauge theory is a non-trivial (cluster) SPT phase. However, it was unclear whether this was a 1D artefact, and moreover, in this 1D case the SPT phenomenology immediately disappears upon explicitly breaking the magnetic symmetry, making it arguably less physically compelling. In the present volume, we find that the Higgs phase is also an SPT phase in higher dimensions, but equally importantly, its physical properties are parametrically robust to breaking the magnetic symmetry. This traces back to the SPT being protected by a higher-form symmetry. In fact, in the prescient Appendix of Ref. [34] it was briefly pointed out that the Higgs phase of $\mathbb{Z}_2$ lattice gauge theory is a higher SPT phase (although its stability and other properties were not investigated). Another impor-

tant symmetry in our story is the global matter symmetry, which is physical (i.e., not a gauge symmetry) on the boundary links. A similar observation was made by Harlow and Ooguri in Ref. [35] where they referred to this as an asymptotic symmetry [36–38], which we discuss in more detail in Sec. 4.4.

An important class of Higgs phases are superconductors, which are charge condensates for a $U(1)$ gauge theory. Ref. [39] has emphasized the topological aspects resulting from the fundamental $U(1)$ charge being fermionic so that the Cooper pair condensate is not in the fundamental representation, leading to a remaining $\mathbb{Z}_2$ topological order in the Higgs phase (see also the related discussion of 'partial Higgs' phases in Ref. [7]). Our proposal that 'Higgs=SPT' concerns the situation of a fundamental charged Higgs boson condensing, which accounts for the salient phenomena of superconductivity, i.e., persistent super-currents and the Josephson effect, in terms of SPT phenomena, as we discuss in detail in Part II [27]. The extra $\mathbb{Z}_2$ features of the fermionic case are readily incorporated in our theory, by sharpening the interpretation of a paired superconductor as a magnetic symmetry-enriched[1] $\mathbb{Z}_2$ topological order (i.e., one might say that more generally 'Higgs=SET' rather than 'Higgs=SPT'). Our interpretation correctly accounts for both the topological order and its interplay with superconductivity, for instance the integer (versus $\mathbb{Z}_2$) valued superconductor vorticity.

## 2 Bird's-eye view of Higgs = SPT

While we have already mentioned SPT phases above, let us briefly recall their precise definition. An SPT phase is the ground state of a symmetric, gapped, nondegenerate Hamiltonian which can be deformed to one with a product state ground state[2] without closing the gap but possibly breaking the symmetry. We say moreover that the SPT is non-trivial if we *must* break the symmetry to connect it to a product state; we can thus say that the symmetry protects certain quantum entanglement in the ground state. A remarkable consequence of this simple criterion is that such phases host zero-energy or gapless edge modes at their boundaries.

The description of the Higgs phase as an SPT phase can be seen as the continuation of a recent re-interpretation of the *deconfined* phase as spontaneously *breaking* a higher-form symmetry. What does this mean more precisely? In the case of the magnetic symmetry of $\mathbb{Z}_2$ gauge theory, it means that while the ground state is an eigenstate of *closed* Wilson loop operators $e^{i \oint A}$ (see Fig. 2(a)), the expectation value of *open* string operators $\langle \mathcal{O}(x)^\dagger e^{i \int_x^y A} \mathcal{O}(y) \rangle$ is vanishingly small (Fig. 2(b)) for any choice of endpoint operator $\mathcal{O}(x)$. This notion is similar to the breaking of usual symmetries. Indeed, the above construction reduces to detecting conventional symmetry breaking in the special case of $d = 1$, where the Wilson loop operator has the dimension of ambient space. In that case, Ising cat states (or the GHZ state, i.e. the superposition of all spins being 'up' and 'down') are still invariant under the global symmetry, whereas acting with the symmetry generator on a finite-but-large region (also called the *two-point functions of the disorder operators*) will orthogonalize the state over that whole region, leading to exponential decay of the open

---

[1]While topological order enriched with the magnetic symmetry has been already investigated in superconductors in [40], SPT properties that originate from the interplay of the magnetic and the global Higgs matter symmetries were not discussed there.

[2]Strictly speaking, this makes sense only in a system with a tensor product Hilbert space and an on-site symmetry action. This tensor product condition actually fails in an exact gauge theory because of the Gauss law constraint (emergent gauge theories like toric code are still completely local however). In that case one will have to be more careful, and work *relatively*, as we will see.

string correlation function[3]. Similar to this $d = 1$ case, one can argue that this (generalized) notion of symmetry breaking can only occur for a long-range entangled quantum state, and serves as an order parameter for the deconfined phase[4].

In contrast, the *Higgs* condensate is the phase which *preserves* this magnetic symmetry. In line with the above discussion, the symmetry-preserving phase should have long-range order (LRO) for the open Wilson string. Because of gauge invariance this can only happen if the endpoint is dressed with a matter field. In other words, the symmetric phase has[5] $\langle \varphi(x)^\dagger e^{i \int_x^y A} \varphi(y) \rangle \to$ const $\neq 0$ at large separations, see Fig. 2(c). This gives invariant meaning to the 'charge condensate' of the Higgs phase. If the matter field $\varphi$ carries charge under some global symmetry, then this phase is a non-trivial SPT. This could be an extra quantum number, like spin or momentum, or something more subtle, such as the relative charge on two sides of an insulating junction, or a flavour symmetry having to do with the emergence of the gauge theory at low energies. We will discuss all these cases in detail throughout this work, and give a flavor of them in Sec. 2.1. The careful treatment of this global symmetry is crucial for deriving properties of the SPT phase. In short, the symmetry is global if and only if there are gauge-invariant charged operators in the spectrum (even if very massive). We will refer to this generally as the matter symmetry, and specify when we have a more concrete setting in mind.

We can here make contact with the picture of the SPT ground state as a condensate of charged disorder operators (i.e. decorated domain walls [28]), constituting of open versions of the symmetry string (or more generally membrane) operator. The open Wilson line above is a disorder operator for the magnetic symmetry, and as an order parameter it matches the string order parameters familiar in 1d SPT phases [44]. We make this connection very explicit in Sec. 3, where we recast the $\mathbb{Z}_2$ Higgs phase in one spatial dimension as the 1D cluster state.

We can thus say that the Higgs condensate is an SPT phase protected by magnetic higher-form symmetry and global matter symmetry. We can also define a magnetic symmetry for $U(1)$ and even certain non-abelian gauge groups [27], and find in all these cases there is a natural SPT structure involving a condensed charge decorating an SPT order parameter. In this sense, the Gauss law is an SPT stabilizer.

## 2.1 Physical implications

What is the physical content of 'Higgs=SPT'? After all, this is an unusual SPT for at least two reasons. First, the global charge symmetry seems suspiciously like a gauge redundancy—such an unphysical symmetry can surely not protect SPT physics. Second, the SPT phase is protected by a higher-form symmetry[6], which appears highly fine-tuned in any lattice model. Let us briefly summarize some key implications, which will address these concerns.

Above, we argued that the string order parameter of the Higgs phase, $\langle \varphi(x)^\dagger e^{i \int_x^y A} \varphi(y) \rangle \neq 0$ can be interpreted as an SPT order parameter if the matter field $\varphi(x)$ is charged un-

---

[3]Note that this conclusion holds for any of the ground states, including the symmetry-broken ones. Here we emphasize the symmetry-preserving cat states, since *even though* they are globally symmetric, the two-point function *still* decays. This is similar to the higher-dimensional case, where the deconfined phase is invariant under closed Wilson loop operators but not open ones.

[4]This has been generalized by Fredenhagen and Marcu beyond the magnetic symmetry preserving line. The closed Wilson loop no longer preserves the ground state but we can divide the open Wilson loop by a closed one of equal length; its vanishing is a robust probe of deconfinement and topological order [41, 42].

[5]To avoid confusion, let us emphasize that we are using the continuum notation for a $\mathbb{Z}_2$ gauge theory. For $U(1)$ gauge theory, one typically works with the Dirac order parameter [43]; see the discussion in Part II [27].

[6]The notable exception being $\mathbb{Z}_n$ ($U(1)$) gauge theories in spatial dimension $d = 1$ ($d = 2$).

der some physical symmetry. A physical symmetry is one which can be probed by local, gauge-invariant charged operators. Just declaring that $\varphi$ is charged is not enough to derive any physical consequences, since it is not gauge-invariant.

One natural route toward a physical matter symmetry is that in the presence of a boundary, one can assign a non-trivial matter charge to the local gauge-invariant Wilson line operators which terminate at the boundary; see Fig. 2(c). Indeed, even though we can intuitively think of a *global* matter symmetry, the presence of a local Gauss law (pinning charge to electric flux) means we can always push it to an effective *boundary* symmetry, which amounts to conserving the electric flux through the edge of the system[7]. A boundary Hamiltonian which respects this matter symmetry as well as the aforementioned magnetic symmetry will have degenerate SPT edge modes, which is a direct consequence of the nonzero vacuum expectation value of the short Wilson line near the boundary. One can say that the matter symmetry is both physical and spontaneously broken at the boundary.

The above general argument can be explicitly verified in the simplest paradigmatic lattice gauge theory: the Fradkin-Shenker model for $\mathbb{Z}_2$ gauge theory in two spatial dimensions. In the case where one includes explicit matter degrees of freedom, we find that for the rough boundary condition (which is the one where the magnetic line symmetry is preserved) the aforementioned boundary electric flux is simply the global parity of the matter degrees of freedom, which is indeed not reducible to a product of local gauge transformations (see Sec. 4). We show that this physical charge symmetry is spontaneously broken (only) at the edge.

The above relied on having magnetic symmetry. Like most higher-form symmetries, this is rather fine-tuned in the phase diagram[8]. Indeed, for the Fradkin-Shenker model, it is only an exact symmetry if $h = 0$ in Fig. 1. Crucially, the *edge mode is generally stable to explicitly breaking the magnetic symmetry*! For this $\mathbb{Z}_2$ lattice gauge theory, we find edge modes in the blue region in Fig. 1(b); see Sec. 4. In summary, while the Higgs and confined phases are adiabatically connected for periodic (or infinite) geometries, the energy gap must close at the edge when there are boundaries[9]. The situation is similar in three spatial dimensions for both $\mathbb{Z}_2$ and $U(1)$ gauge theory. In contrast, for $\mathbb{Z}_2$ gauge theory in 1D (or $U(1)$ in 2D) the magnetic symmetry is a 0-form symmetry, in which case the edge mode is unstable [33].

The above discussion envisions a gauge theory with a hard edge, which is a rather subtle (if not contrived) concept. In SPT physics, one can commonly re-interpret edge modes as arising at a spatial interface from the SPT phase to a trivial phase. However, such a trivial phase cannot be realized within the gauge theory itself, which does not allow for product states. One setting where this viewpoint does apply naturally is in an *emergent* gauge theory, where, say, the Fradkin-Shenker model arises energetically as a low-energy theory in a tensor product Hilbert space. This is enough to give rise to an SPT phase with protected edge modes, and in this case, there is a well-defined notion of a product state which is separated from the Higgs phase by a bulk phase transition.

If one wants to realize interface modes with an *exact* gauge symmetry, then one can instead consider a theory with *multiple* flavors of gauge charged matter. The *relative* charge is physical: it acts on a short Wilson line with different matter fields on each end. The Higgs phases where different matter fields condense are distinguished by these relative charges. In particular, they are separated by a bulk SPT transition, and a spatial interface

---

[7]A related perspective offered in Ref. [33] is that while the act of gauging trivializes a global symmetry in the bulk, in the presence of a boundary the symmetry can survive and acts at the edge.

[8]Note that we are talking about *exact* (higher) symmetries, as we discuss in detail in lattice models (Sec. 4), and not emergent ones.

[9]Unless of course one explicitly breaks global matter symmetry—which is indeed in principle possible by virtue of it being a physical symmetry.

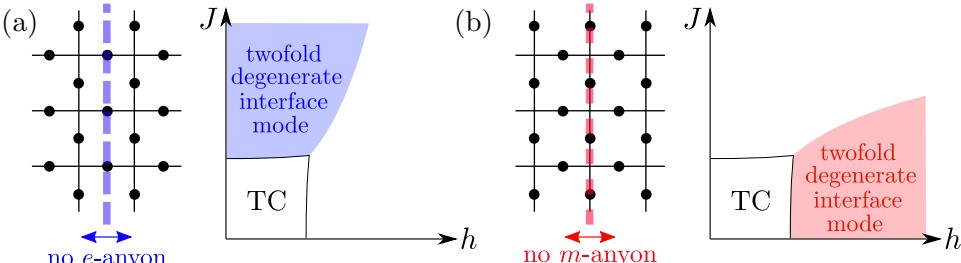

Figure 3: **Implications of Higgs=SPT for toric code in a field.** We can observe edge (more precisely, interface) modes of the Higgs phase even for lattice gauge theories without explicit gauge-invariant matter degrees of freedom. For instance, consider the toric code in a field [5, 6, 45–47], $H = -\sum_v A_v - \sum_p B_p - h \sum_l \sigma_l^x - J \sum_l \sigma_l^z$. (a) The fact that the matter 'symmetry' is a gauge symmetry is self-evident as there are not even any matter degrees of freedom on which to act. However, one way to introduce a physical matter symmetry is by imitating the SIS junction of superconductors: we introduce a line defect (dashed line) across which charge (i.e., an $e$-anyon) is not allowed to move or tunnel. The physical symmetry is then the relative charge of the two regions, i.e., the conservation of electric flux $\prod_{l\in\text{defect}} \sigma_l^x$. The fact that the Higgs condensate (i.e., large $J$) is an SPT then manifests itself into an interface mode localized to this defect line (see Sec. 3.2.3 and Sec. 4.5). We sketch the phase diagram where this leads to a twofold degeneracy (blue shaded region). (b) Similarly, introducing an insulating strip (dashed line) across which $m$-anyons are not allowed to move gives rise to interface modes deep in the confined phase (red region). Indeed, by electro-magnetic duality, we can say 'confined=SPT' protected by higher-form electric symmetry and global 'magnetic matter' symmetry.

between the two carries edge modes. Neither is really trivial, but we can say they *differ* by a well-defined SPT phase.

Even within a *single* Higgs phase in an exact gauge theory, we can define an interesting interface by studying an insulating junction across which gauge charge hopping is forbidden. This simulates the Higgs-Higgs' junction above, where the matter symmetry is the relative gauge charge between the two sides of the junction, or equivalently the electric flux through it. We find localized modes protected by this symmetry and the magnetic symmetry. These results even give new insights into as basic a model as the toric code in a field, see Fig. 3. Moreover, in the $U(1)$ case, tunnelling weakly breaks this symmetry, leading to a current which gives an SPT interpretation of the Josephson effect at SIS interfaces. In Part II [27], we also study the interface between the $U(1)$ Higgs and magnetic symmetry breaking state, which describes a superconductor in the Coulomb vacuum, and find that supercurrents are also a consequence of the SPT, although there are no edge modes as such.

## 2.2    Roadmap to this work

The remainder of this work is structured as follows. In Sec. 3, we discuss a web of analogies between $\mathbb{Z}_2$ gauge theory and SPT phases in the minimal 1+1d setting. The Higgs phase of a $\mathbb{Z}_2$ gauge theory is related to the cluster SPT chain [33], with the Gauss law playing the role of an SPT stabilizer. We discuss the zero modes appearing at boundaries and interfaces, the relation to 'hidden symmetry breaking', and what happens when the magnetic symmetry is broken. This section is meant to untangle all the conceptual subtlety of the Higgs=SPT phenomena in a tractable setting.

In Sec. 4, we turn our attention to the main example, the 2+1d Fradkin-Shenker model,

which we introduce in Sec. 4.1. To see 'Higgs=SPT', we add explicit matter to the model, which is charged under a global $\mathbb{Z}_2$ symmetry. In the absence of magnetic fluctuations $h = 0$, we can detect a $\mathbb{Z}_2 \times \mathbb{Z}_2[1]$ ([$p$] denotes $p$-form symmetry) SPT using the string order parameter / open Wilson line (Sec. 4.2). After reviewing the well-known phase diagram with periodic boundary conditions (Sec. 4.3), we use these string operators to argue that the global $\mathbb{Z}_2$ symmetry is spontaneously broken at the "rough" boundary (Sec. 4.4). The "smooth" boundary meanwhile is an interface to a confined phase and does not preserve the magnetic symmetry (unless we do not enforce boundary Gauss laws as in Appendix B). We study the model for $h > 0$ both numerically and in a solvable limit, and in both cases we find that the edge mode persists in a large region of the phase diagram, as shown in Fig. 1(b).

In addition, Sec. 4.5 highlights how the physical matter symmetry of the model can arise in a variety of situations, e.g., from a two-Higgs model, an emergent gauge theory, or in the context of a 'SIS interface' familiar from superconductors. This also leads to scenarios with bulk SPT phase transitions as discussed in Sec. 4.6. After briefly touching upon higher-dimensional generalizations (Sec. 4.7), we present a deeper perspective on 'Higgs=SPT' by analysing the defining boundary anomaly (Sec. 4.8)—this also forms the jumping off point for the case with continuous gauge symmetries in Part II [27].

Finally, Sec. 5 aims to export the insights obtained from 'Higgs=SPT' in order to apply them to more general SPT phases, even those entirely unrelated to gauge or Higgs phases. In particular, Sec. 5.1 shows how the 'SIS junction' of Higgs phases provides a novel way to probe SPT phases via a bulk-defect correspondence. Secondly, Sec. 5.2 makes the case that SPT phases protected by higher-form symmetries have properties which are parametrically robust to explicit breaking of those symmetries.

# 3 How gauge theory can arise from living in an SPT model: a thought experiment in 1+1D

Here we discuss a thought experiment which illustrates how living in a symmetry-protected topological (SPT) phase can in certain cases be indistinguishable from living in a gauge theory. We first review the cluster SPT chain in Sec. 3.1, after which we explore how it becomes a gauge theory in a certain limit in Sec. 3.2. In this section, we restrict to 1+1d (i.e., one spatial dimension) and $\mathbb{Z}_2$ symmetry. We will see that *even this elementary setting* already showcases many of the main messages of this work. While the content of this section largely follows Ref. [33], the narrative as well as some technical aspects—such as the SIS interface—are novel.

## 3.1 Alice in spin chain: trivial and SPT phases

Suppose Alice lives in a spin-1/2 chain. We take the Hilbert space to be $\mathcal{H} = \mathcal{H}_{\text{vertices}} \otimes \mathcal{H}_{\text{link}}$, with spins living on 'vertices' (labeled by an integer $n$) with Pauli operators $X_n, Y_n, Z_n$, and other spins living on 'links' (labeled by a half-integer $n + \frac{1}{2}$) with Pauli operators $\sigma_{n+1/2}^{x,y,z}$. One choice of Hamiltonian for Alice's universe is the cluster model[10] [48]:

$$H_{\text{cluster}} = -\sum_n \sigma_{n-1/2}^x X_n \sigma_{n+1/2}^x - \sum_n Z_n \sigma_{n+1/2}^z Z_{n+1}. \tag{1}$$

---

[10]This differs from the usual definition by a Hadamard transformation $\sigma^x \leftrightarrow \sigma^z$ on the link qubits; we do this to make the analogy with the usual Gauss law clearer.

This is a stabilizer model, meaning all terms commute and the ground state is defined by minimizing each term individually. As we will discuss, it defines a non-trivial SPT phase protected by a $\mathbb{Z}_2 \times \mathbb{Z}_2$ symmetry [49] generated by

$$P = \prod_n X_n \qquad \text{and} \qquad W = \prod_n \sigma^z_{n+1/2}. \qquad (2)$$

While unbroken in the ground state, these symmetries distinguish the above ground state from a trivial one, such as the ground state of $H_{\text{triv}} = -\sum_n \left( X_n + \sigma^z_{n+1/2} \right)$ (note that this respects the same $\mathbb{Z}_2 \times \mathbb{Z}_2$ symmetries). If Alice lived in a universe with a Hamiltonian $H = \lambda H_{\text{cluster}} + H_{\text{triv}}$ with tuning parameter $\lambda$, she would see that the small-$\lambda$ and large-$\lambda$ regimes are separated by a phase transition. (In particular, the correlation length diverges at the free boson quantum critical point at $\lambda = 1$ [50,51].) Having a short-range entangled[11] phase which cannot be smoothly connected to the product state in the presence of symmetries is the defining characteristic of a (non-trivial) SPT phase [52,53].

### 3.1.1 Nonlocal order parameters

How would Alice distinguish the two different phases? Neither phase spontaneously breaks the symmetry, so there is no local order parameter. However, she would uncover that there exists a *nonlocal* order parameter. For instance, the small-$\lambda$ (trivial) phase has long-range order for the string operator $\prod_{i \leq n < j} \sigma^z_{n+1/2}$, i.e., the $W$ symmetry operator on a large-but-finite region. In contrast, in the large-$\lambda$ phase, we have that it vanishes exponentially fast:

$$\left\langle \prod_{i \leq n < j} \sigma^z_{n+1/2} \right\rangle \sim e^{-|i-j|/\xi} \qquad (\text{if } |\lambda| > 1). \qquad (3)$$

This is similar to what we expect for a phase that spontaneously breaks $W$. However, the situation is reversed for the decorated domain wall operator, $Z_i \prod_{i \leq n < j} \sigma^z_{n+1/2} Z_j$, which has long-range order in the SPT phase, and vanishes in the trivial phase [54]. The fact that we have long-range order for a $W$-string *only if it is decorated by an endpoint operator which is odd under $P$* gives a discrete ('topological') invariant distinguishing the SPT phase from the trivial phase[12] [44].

### 3.1.2 Edge modes

A characteristic feature of a nontrivial SPT phases is the emergence of edge modes, which are protected as long as symmetries are preserved. These either correspond to ground state degeneracies (i.e., spontaneous symmetry breaking at the edge) or gapless edge modes (celebrated for topological insulators in two and three spatial dimensions [9,11,55,56]). It is straightforward to show[13] that terminating the above chain gives rise to exponentially localized zero-energy operators if $|\lambda| > 1$. More generally, the existence of such zero modes can be derived from the long-range order of the string order parameter, as demonstrated in Fig. 4(a). In fact, even if Alice's universe had no boundary (or if she were not able

---

[11]The two Hamiltonians are related by the finite-depth circuit $H \times \prod CZ \times H$ where $H$ is a Hadamard transformation on all the link qubits [48].

[12]Indeed, more generally it can be argued that any phase which is gapped and symmetric under $W$, must have long-range order for *some* endpoint decoration of its corresponding domain wall operator; the symmetry properties of the endpoint operator then serve as a topological invariant [44].

[13]E.g., if we terminate the system with a link qubit, then we have two localized anti-commuting Pauli operators which commute with the Hamiltonian: $X_L = \sigma^x_{1/2}$ an $Z_L = \frac{1}{\sqrt{1-1/\lambda^2}} \left( \sigma^z_{1/2} Z_1 + \lambda^{-1} \sigma^z_{1/2} X_1 \sigma^z_{3/2} Z_2 + \cdots + \lambda^{-n} \sigma^z_{1/2} \left( \prod_{i=1}^n X_i \sigma^z_{i+1/2} \right) Z_{n+1} + \cdots \right)$ for $|\lambda| > 1$.

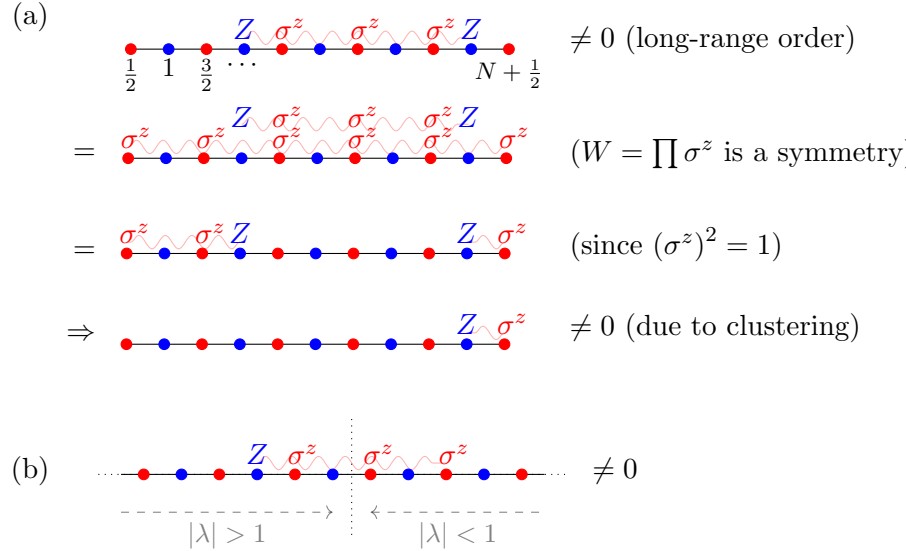

Figure 4: **Edge modes from SPT string order.** (a) Long-range order of a non-trivial SPT string order parameter implies edge modes for a finite chain. The edge operator anticommutes with $P$ in Eq. (2), implying a degeneracy. (b) A spatial interface between the SPT and trivial phase also carries a zero-energy mode by virtue of the mixed string order parameter having long-range order.

to travel there), the edge mode could also be observed if we had a spatially-dependent coupling $\lambda(x)$ such that we could have a spatial interface between the trivial and SPT phase, as shown in Fig. 4(b).

## 3.2    Bob in constrained spin chain: emergence of gauge theory

Bob lives in a more restricted part of Alice's universe. If we consider Eq. (1) with different prefactors $K$ and $J$ for the two SPT stabilizers,

$$H = -K \sum_n \sigma^x_{n-1/2} X_n \sigma^x_{n+1/2} - J \sum_n Z_n \sigma^z_{n+1/2} Z_{n+1} - \sum_n X_n, \qquad (4)$$

then in Bob's universe $K \to \infty$, i.e., it is larger than any energy scale Bob can ever have access to (for related discussions of such a Hamiltonian, see e.g. Refs. [33, 57–59]). This means that a short-range entangled phase must be in the non-trivial SPT phase. To see this, note that large $K$ pins $\sigma^x_{n-1/2} X_n \sigma^x_{n+1/2} = 1$. Products of these inescapably imply long-range order in the non-trivial string order parameter $\sigma^x_{i-1/2} \prod_{i \leq n \leq j} X_n \sigma^x_{j+1/2} = 1$, its endpoints charged under $W$ in Eq. (2). The only way of escaping the SPT phase is by breaking the $W$ symmetry of the ground state: for example, at small $|J|$, the symmetry $W$ is broken spontaneously by long-range order[14] in $\sigma^x_{n+1/2}$. Alternatively, one can break $W$ *explicitly* which will be discussed in Sec. 3.2.8.

This gives an ironic twist of fate: since $K \to \infty$ pins one of the SPT stabilizers, we have no trivial phase to compare to. This makes it harder to even notice that this universe has a non-trivial SPT phase, since we cannot tune a bulk phase transition between the two as in Alice's universe! Let us discus how Bob might go about analyzing the universe he finds himself in (Sec. 3.2.1). We will see how it is still possible to observe the SPT

---

[14]One way to see this is that $J = 0$ implies $X_n = 1$ (in the ground state), which can be combined with the aforementioned string order parameter; also see Sec. 3.2.7.

properties (Sec. 3.2.2 and Sec. 3.2.3) and how one might initially misinterpret the nature of this phase (Sec. 3.2.4 and Sec. 3.2.5).

### 3.2.1 SPT stabilizer as Gauss law

If Bob lives in a universe described by Eq. (4) with $K \to \infty$, he would say that the Hamiltonian is [57]

$$H = -J \sum_n Z_n \sigma_{n+1/2}^z Z_{n+1} - \sum_n X_n , \tag{5}$$

with the condition that we only consider states for which

$$G_n = \sigma_{n-1/2}^x X_n \sigma_{n+1/2}^x = 1 . \tag{6}$$

Bob would conclude that we are actually using a redundant description, with the physical Hilbert space being

$$\mathcal{H}_{\text{phys}} = \{|\psi\rangle \in \mathcal{H} \quad \text{with } G_n|\psi\rangle = |\psi\rangle\} \quad \subset \quad \mathcal{H}. \tag{7}$$

From Bob's point of view, *his universe is indistinguishable from a conventional (lattice) gauge theory*, where $G_n = 1$ is a Gauss law. Equivalently, we can say $G_n$ generates a gauge transformation

$$Z_n \to s_n Z_n \quad \text{and} \quad s_n \sigma_{n+1/2}^z s_{n+1} \quad \text{with } s_n \in \{\pm 1\}. \tag{8}$$

Rather than using only gauge-invariant states, Bob finds it sometimes convenient to still use the redundant Hilbert space $\mathcal{H}$ (however see Sec. 3.2.5), whilst reminding himself that at the end of the day he should only ask questions about gauge-invariant observables.

In light of Eq. (8), we can refer to the site variables as 'matter' and the link variables as the 'gauge field'. In fact, writing $\sigma^z = e^{iA}$ (with $A \in \{0, \pi\}$), we can identify $W = \prod_n \sigma_{n+1/2}^z$ with the Wilson loop $e^{i \oint A}$ measuring the magnetic flux threading Bob's ring universe; this is hence also called the *magnetic symmetry*. In contrast, $P = \prod_n X_n$ seems like an unphysical symmetry, since the Gauss law dictates that $P = \prod_n G_n = 1$—at least for periodic boundary conditions. We now discuss three routes by which a physical symmetry can be associated with the gauge charges (Sec. 3.2.2, Sec. 3.2.3 and Sec. 3.2.6).

### 3.2.2 First manifestation: zero modes at the edge of the universe

If Bob's universe had a boundary, then he would find a zero-energy edge mode in the large-$J$ phase. This is simplest in the setting where we end Bob's universe with link variables, such that the Gauss operators are well-defined for all sites $n$. The crucial point is that the global charge is no longer pinned by the Gauss law, i.e., $P = \prod_n X_n \neq \prod_n G_n = 1$. Instead we find that it acts on the boundary links [33]:

$$1 = \prod_n G_n = \sigma_{1/2}^x X_1 X_2 \cdots X_N \sigma_{N+1/2}^x \quad \Rightarrow \quad P = \sigma_{1/2}^x \sigma_{N+1/2}^x. \tag{9}$$

The action of $P$ is thus not purely 'gauge' in the presence of boundaries. If any local Hamiltonian commutes with Eq. (9), it must also commute with $\sigma_{1/2}^x$ and $\sigma_{N+1/2}^x$ individually. Since these anticommute with the $W$ symmetry, we conclude that in the presence of boundaries, the ground state must be degenerate. Moreover, if the ground state is short-range entangled, we conclude we have (at least) a twofold degeneracy per edge. (This is straightforward from the SPT perspective as discussed in Sec. 3.1.2.)

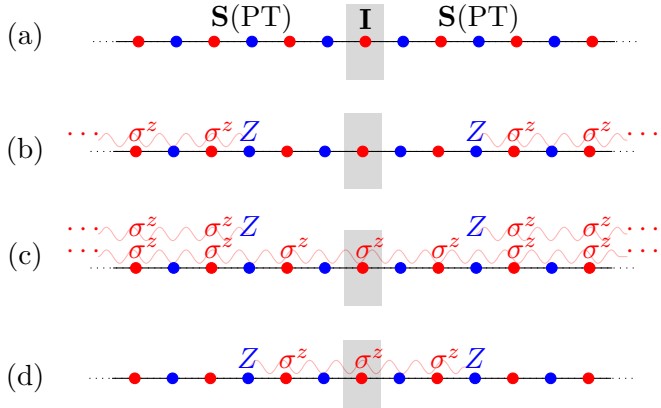

Figure 5: **Edge modes at an insulating region.** (a) We consider the SPT (i.e., Higgs) phase (**S**) with an insulating (**I**) region inserted (gray), preventing charge from tunneling through. (b) In the SPT, an open Wilson string (red) leaves the ground state invariant if the ends are dressed with a $P$-odd operator (blue; for the fixed-point SPT this is a pauli-$Z$). (c) The Hamiltonian commutes with magnetic symmetry $W$, which we can thus freely insert (red line). (d) Using $(\sigma^z)^2 = 1$, we obtain that a (dressed) Wilson string extending across the insulating region must commute with the Hamiltonian (see Eq. (11) for the operator which commutes even away from the fixed-point limit), which anti-commutes with $P_L$ (i.e., the conserved charge to the left of the insulating region), implying a two-fold degeneracy.

Note that the above argument only relied on the Hamiltonian having $P$ and $W$ symmetry (together with the Gauss law which we used to derive Eq. (9)). Hence, it also applies to the small-$J$ phase. In Sec. 3.2.7 we discuss that this regime spontaneously breaks $W$. It thus has a symmetry-breaking degeneracy—both for periodic and open boundaries (in higher dimensions this is the deconfined phase of the gauge theory with topological degeneracy). In contrast, for large-$J$ the ground state is unique with periodic boundary conditions. It is sometimes called the *Higgs phase* since charges are condensed: indeed that is exactly our SPT order parameter $\langle Z_n \prod_{n \leq i < m} \sigma^z_{i+1/2} Z_m \rangle \neq 0$. Field-theoretically, $\langle \psi(x)^\dagger e^{i \int_x^y A(r) \mathrm{d}r} \psi(y) \rangle \neq 0$. Note that the geometry with boundaries thus has a ground state degeneracy both in the small- and large-$J$ phases. In fact, this degeneracy persists at the critical point between these two phases—it is an example of a topologically non-trivial symmetry-enriched CFT [60].

### 3.2.3 Second manifestation: zero modes at an SIS junction

Suppose Bob's universe has no boundary (or he cannot travel there). How can he detect that the Higgs phase is a non-trivial SPT phase? This seems subtle since the SPT is protected by $\prod_n X_n$, and from Bob's perspective this is indistinguishable from the gauge symmetry in the bulk of the system. One way of making this symmetry physical is by talking about *relative* charge. Bob's good friend Brian J. recommends him to consider a tiny insulating region separating two Higgs regimes in space. I.e., we impose a constraint that charges cannot tunnel/hop across, say, the bond at $n_0 + 1/2$. By definition, this means that the charge $P_L = \prod_{n \leq n_0} X_n$ is a symmetry. Concretely, this can be achieved by setting the charge-hopping $J = 0$ across that bond: in this case the Hamiltonian is

the same as before, up to removing the term $Z_{n_0}\sigma^z_{n_0+1/2}Z_{n_0+1}$. Note that $P_L$ is physical[15] (e.g., $Z_{n_0}\sigma^z_{n_0+1/2}Z_{n_0+1}$ is a gauge-invariant local operator which is charged under $P_L$). Similar to the argument in Sec. 3.1.2, the nontrivial SPT string order parameter implies that such an SPT-insulator-SPT (SIS) junction must carry a zero-energy SPT mode[16]! In fact, one can construct an explicit zero-energy qubit for the above model:

$$X_{\text{SIS}} = P_L = \prod_{n \leq n_0} X_n = \sigma^x_{n_0+1/2}, \tag{10}$$

$$Z_{\text{SIS}} = \frac{1}{(1-1/J)^2} \sum_{a=-\infty}^{n_0} \sum_{b=n_0+1}^{\infty} \frac{1}{J^{b-a-1}} Z_a \sigma^x_{a+1/2} \times \prod_{c=a}^{b-1} \sigma^z_{c+1/2} \times \sigma^x_{b-1/2} Z_b. \tag{11}$$

These two anti-commuting Pauli operators are exponentially localized around $n \approx n_0$ and commute with the Hamiltonian[17]. To gain some insight into this unsightly expression for $Z_{\text{SIS}}$, let us observe that in the limit $|J| \to \infty$ it reduces to $Z_{n_0}\sigma^z_{n_0+1/2}Z_{n_0+1}$. The above expression (11) simply endows this with some exponential tails such that it remains an exact zero-energy mode which is well-defined (i.e., normalizable) for $|J| > 1$ (i.e., the Higgs phase). The intuition for this zero-mode is sketched in Fig. 5.

   This points to a general and novel way of probing the non-trivial edge mode of generic SPT phases, even in the absence of a hard edge or in absence of a nearby trivial phase. This is explored further in Sec. 5.1. In the particular case of $U(1)$ gauge theory in higher dimensions, where the Higgs phase is a superconductor, the SIS junction supports a $U(1)$ superfluid. In that case, adding perturbatively small tunneling across the junction induces the famed Josephson effect! This perspective thus re-interprets and generalizes the Josephson junction as an SPT phenomenon, which we explore further in Part II [27].

### 3.2.4   Gauge-fixing: Higgs phase as symmetry-breaking?

There are thus multiple ways of seeing that the Higgs phase is a non-trivial SPT phase. However, there are several ways in which Bob might be misled about the nature of this phase. Firstly, he might suspect that it has to do with symmetry-breaking, since $Z_n\sigma^z_{n+1/2}Z_{n+1}$ looks like an Ising interaction for the matter fields with minimal coupling to the gauge field. In an effort to make this precise, Bob realizes that he can use the Gauss law to effectively get rid of the gauge field. To this end, he introduces the notion of gauge-fixing, which corresponds to choosing for each physical state $|\psi\rangle \in \mathcal{H}_{\text{phys}}$ an unphysical representative, e.g. $|\psi_{\text{fix}}\rangle = \prod_n \left(1 + \sigma^z_{n+1/2}\right)|\psi\rangle \in \mathcal{H}$, whose gauge orbit[18] is $|\psi\rangle$. This state satisfies the gauge-dependent condition $\sigma^z_{n+1/2} = 1$, effectively removing any gauge field dynamics. The action of the Hamiltonian (5) on the remaining degrees of freedom is then:

$$H_{\text{fix}} = -J \sum_n Z_n Z_{n+1} - \sum_n X_n. \tag{12}$$

   At face value, this seems to suggest the large-$J$ phase spontaneously breaks $P = \prod_n X_n$. However, this is of course misleading. The long-range order of $\langle Z_n Z_m \rangle$ in this gauge is just a disguised form of the non-trivial SPT string order we encountered in Sec. 3.1.1. Such a string order does not imply any bulk degeneracy, and indeed Eq. (12)

---

[15]Gauge symmetry does imply that $P_R = \prod_{n \geq n_0} X_n = P_L P = P_L$.

[16]To see this, note that by the Gauss law $P_L = \sigma^x_{n_0+1/2}$. This symmetry anticommutes with $W = \prod_n \sigma^z_{n+1/2}$.

[17]Here we have simplified the expression for $Z_{\text{SIS}}$ using the Gauss law $G_n = 1$.

[18]The map can be inverted by gauge-symmetrizing: $\prod_n (1 + G_n)|\psi_{\text{fix}}\rangle \propto |\psi\rangle$ (ignoring boundary/global conditions).

is still constrained by the global condition $P = \prod_n X_n = \prod_n G_n = 1$, at least for periodic boundary conditions. The ground state is only degenerate in the presence of boundaries (see Sec. 3.1.2), but that is localized to the edge (i.e., the degenerate ground states are indistinguishable in the bulk).

### 3.2.5   No need to keep track of matter: Higgs phase as product state?

A second source of misinterpreting the SPT nature of the Higgs phase is that the Gauss law seems to make matter a dummy variable. Indeed, $X_n = \sigma^x_{n-1/2}\sigma^x_{n+1/2}$ implies we can determine the matter configuration from the link variables. Similar to above, one could gauge-fix to get rid of matter. Here we take a more elegant (but equivalent) route: there exists a finite-depth unitary which disentangles matter and gauge field[19]. More precisely, if we perform the following unitary transformation $U$, composed of a Hadamard transformation on the link qubits (i.e., $H = \frac{\sigma^x + \sigma^z}{\sqrt{2}}$), followed by a $CZ$ (Controlled-$Z$) on every bond, and Hadamard again, then the Gauss operator is mapped to $G_n = X_n$. Thus, the Gauss law $G_n = 1$ freezes out matter, and Eq. (5) becomes

$$UHU = -J \sum_n \sigma^z_{n+1/2} - \sum_n \sigma^x_{n-1/2}\sigma^x_{n+1/2}, \tag{13}$$

with no remaining Gauss law.

The large-$J$ phase is now simply a product state! However, it is important to recognize that it is the fate of *any* SPT to be a finite-depth circuit away from triviality. Note that this is only possible using circuits whose gates do *not* commute with the symmetry (and indeed $CZ$ does not commute with $P$). Such a mapping obfuscates the physical properties that make SPT phases interesting. Firstly, the mapping is not on-site and thus breaks down in the presence of boundaries; indeed, we know the SPT phase has degeneracies with open boundaries, whereas the product state does not! Secondly, even in the absence of boundaries, in Sec. 3.2.3 we saw the Higgs phase has localized zero modes for an SIS junction; this is easier to overlook after having disentangled matter and gauge field (or equivalently, in the unitary gauge). In this case, the SIS junction corresponds to preserving $P_L = \prod_{n \le n_0} X_n = \sigma^x_{n_0+1/2}$, and on first sight this local symmetry $\sigma^x_{n_0+1/2}$ might look rather strange, but one can observe that this can be interpreted as conserving the electric flux through the bipartition, which does physicaly capture the 'no-tunneling' defect. Indeed Bob would find that this leads to a twofold ground state degeneracy in the Higgs ground state, since this electric flux anticommutes with the global magnetic symmetry. See Fig. 3 for a higher-dimensional analog. (In Part II, we generalize this to the case for $U(1)$ gauge theory and SIS junctions, and the related Josephson effect [27].)

Gauge-fixing of this kind that eliminates the matter fields can thus be dangerous or misleading when the key properties that concern us relate to SPT physics—although it is fine if care is taken with defining the relevant symmetries in the gauge-fixed language.

### 3.2.6   Third manifestation: multiple Higgs phases and bulk SPT transitions

The SPT nature of the Higgs phase becomes more apparent if there are *multiple* choices of Higgs condensates. In such cases, they will be separated by a bulk SPT transition. Moreover, the spatial interface between distinct Higgs phases will carry localized edge modes (even without inserting an insulating region as in Sec. 3.2.3).

In 2+1D case we will discuss how this scenario naturally arises if we have more than one Higgs field, where the physical symmetry will be the *relative* charge of the two fields

---

[19]The net result is equivalent to expressing our Hamiltonian in terms of gauge-invariant operators, see Appendix C of Ref. [33].

(Sec. 4.6.2). In our current 1+1D setting, we mention an even simpler scenario which relies on the Hamiltonian being real (i.e., it commutes with a time-reversal symmetry $T = K$ which is simply complex conjugation): consider an extension of Eq. (5) given by

$$H_{\text{ext}} = -J \sum_n Z_n \sigma^z_{n+1/2} Z_{n+1} - J' \sum_n Y_n \sigma^z_{n+1/2} Y_{n+1} - \sum_n X_n \qquad (14)$$

with the same Gauss law (6) as before. The large-$J$ and large-$J'$ phases are both Higgs condensates, having long-range order for a gauge-invariant matter-charged operator:

$$\langle \mathcal{O}^\dagger_m \sigma^z_{m+1/2} \sigma^z_{m+3/2} \cdots \sigma^z_{n-1/2} \mathcal{O}_n \rangle \neq 0 \quad \text{where } P\mathcal{O}_n P = -\mathcal{O}_n. \qquad (15)$$

However, they are separated by a quantum phase transition since the endpoint of the two SPT string order parameters is real in one case and imaginary in the other: $\mathcal{O}_n = Z_n$ and $\mathcal{O}_n = Y_n$ have different $T$-charges. This particular SPT transition has a central charge $c = 1$ (for $|J| = |J'| > \frac{1}{2}$). Repeating the disentangling procedure in Sec. 3.2.5 gives

$$U H_{\text{ext}} U = J' \sum_n \sigma^x_{n-1/2} \sigma^z_{n+1/2} \sigma^x_{n+3/2} - J \sum_n \sigma^z_{n+1/2} - \sum_n \sigma^x_{n-1/2} \sigma^x_{n+1/2}. \qquad (16)$$

Even in this case, it is clear that large-$J'$ is a non-trivial SPT phase, but in the original basis (Eq. (15)) the $J$ and $J'$ phases are on equal footing (i.e., related by an on-site change of basis). One can say that the two Higgs phases *differ* by an SPT phase.

### 3.2.7 The deconfined phase

Thus far we have focused on the Higgs phase of the gauge theory. This is indeed the object of study in the present work. To make the analogy with the higher-dimensional cases clearer, let us briefly comment on the small-$J$ phase. Eq. (13) shows this phase spontaneously breaks the magnetic symmetry $W$, where Wilson strings will have exponentially-decaying correlations. This, in a sense to be made precise below, corresponds to the deconfined phase of the gauge theory. Indeed, in 1+1D, recall that the symmetry-breaking phase has deconfined domain wall excitations, in this case corresponding to our (gauge-invariant) charge operators $\cdots \sigma^z_{n-3/2} \sigma^z_{n-1/2} Z_n$ (in the basis of Eq. (5)). More generally, in a deconfined phase, gauge charges are not condensed in the ground state, but they can nevertheless move freely after being created. On moving to higher dimensions, the magnetic symmetry becomes a higher-form symmetry such that its spontaneous breaking gives a ground state degeneracy which depends on the genus of the manifold—reproducing the celebrated phenomenon of topological degeneracy.

### 3.2.8 Explicitly breaking magnetic symmetry

We have only considered Hamiltonians which commute with the magnetic symmetry $W = \prod_n \sigma^z_{n+1/2}$. We could add terms which explicitly break this, like $h \sum_n \sigma^x_{n+1/2}$. This has quite drastic consequences in the 1+1D case. In particular, the SPT phase is destroyed and the zero-energy edge mode is immediately gapped out [33]. However, this a low-dimensional artifact. As we will discuss in Sec. 4, in higher dimensions the edge mode (including the one at the SIS interface) remains parametrically *stable* upon violating the magnetic symmetry[20], owing to the robustness of higher-form symmetries. Similarly, the deconfined phase (Sec. 3.2.7) becomes immediately confined in 1+1D, since spontaneous symmetry-breaking of a global symmetry is not stable against explicitly breaking the symmetry[21]. In contrast, spontaneously breaking higher symmetries is robust to explicit symmetry breaking, which indeed underlies the notion of intrinsic topological order [19,20].

---

[20]More precisely, for $\mathbb{Z}_2$ ($U(1)$) gauge theories it is robust in dimensions $2 + 1D$ ($3 + 1D$) and higher.

[21]However, see the discussion in Ref. [33] where translation symmetry can stabilize the deconfined phase.

# 4  $\mathbb{Z}_2$ Lattice Gauge Theory

Let us now study the "Higgs=SPT" phenomenon in $\mathbb{Z}_2$ gauge theory. For clarity and conceptual simplicity, we mostly focus on lattice gauge theory in two spatial dimensions, although we also discuss generalizations.

In the previous section, we saw how an SPT model (in a tensor product Hilbert space) could become indistinguishable from a gauge theory in a particular limit. In this section, we start by studying the gauge theory outright, which we review in Sec. 4.1, and we do not presume that it secretly arises as the low-energy theory of an SPT model. In Sec. 4.2 we show how deep in the Higgs phase, the lattice gauge theory Hamiltonian is essentially that of the 2D cluster phase on the Lieb lattice, which is an SPT phase protected by matter symmetry as well as a 1-form (magnetic) symmetry. After reviewing the well-known Fradkin-Shenker phase diagram with periodic boundary conditions (Sec. 4.3), we show how boundary conditions which respect the aforementioned symmetry lead to edge modes (Sec. 4.4)—separating the Higgs and confined regimes by a boundary phase transition!

Since studying gauge theories with hard boundaries is rather subtle, Sec. 4.5 shows how the Higgs=SPT phenomenon can also be observed in the bulk upon introducing a defect line. In fact, one can even find bulk SPT transitions by slightly generalizing the aforementioned lattice gauge theory set-up: either one considers the Gauss law to be emergent (Sec. 4.6.1) or one introduces a second Higgs field (Sec. 4.6.2); this also gives a re-interpretation of the aforementioned edge modes as actually being *interface* modes between distint SPT phases (Sec. 4.6.3). These alternative perspectives underline the physical relevance of 'Higgs=SPT', even without studying hard boundaries. In Sec. 4.7 we comment on how the above results generalize to higher dimensions.

Finally, Sec. 4.8 shows how the symmetries of the Higgs phase act anomalously on its boundary, making a direct link to the classification of SPT phases. This rephrases the Higgs=SPT phenomenon in a more general language, which also forms a jumping-off point for studying other (continuous) gauge groups, which are discussed in Part II [27].

## 4.1  Fradkin-Shenker model in $2+1D$ and its symmetries

We consider spin-1/2's (i.e., qubits) on the links and vertices of the square lattice. We write $X_v, Y_v, Z_v$ for Pauli matrices on a vertex $v$ and $\sigma_l^{x,y,z}$ for Pauli matrices on a link $l$. Sometimes we may write $\sigma_{v,v'}^{x,y,z}$ to denote the link connecting two neighboring vertices $v$ and $v'$. It will be convenient to introduce star and plaquette terms [6]:

$$A_v = \prod_{l\in v} \sigma_l^x = \begin{array}{c} \sigma^x \\ \sigma^x \!\!\!- \!\!\! \sigma^x \\ \sigma^x \end{array} \qquad \text{and} \qquad B_p = \prod_{l\in p} \sigma_l^z = \begin{array}{c} \sigma^z \\ \sigma^z \quad \sigma^z \\ \sigma^z \end{array} \ . \tag{17}$$

For every vertex we enforce the Gauss law $G_v = 1$ in the whole Hilbert space where $G_v = X_v A_v$, i.e.:

$$G_v = \begin{array}{c} \sigma^x \\ \sigma^x\, X\, \sigma^x \\ \sigma^x \end{array} = 1 \ . \tag{18}$$

Then the Hamiltonian for $\mathbb{Z}_2$ lattice gauge theory coupled to Ising matter, known as the Fradkin-Shenker model [7], is

$$H = -\sum_v X_v - \sum_p B_p - J \sum_{\langle v,v'\rangle} Z_v \sigma_{v,v'}^z Z_{v'} - h \sum_l \sigma_l^x \ . \tag{19}$$

The first term can be interpreted as a chemical potential for matter; note that by the Gauss law (18) this can equivalently be written as $\sum_v A_v$. The second term measures magnetic flux. The first two terms commute and stabilize the deconfined phase. The $J$-term introduces dynamics to the matter fields; similarly the $h$-term does so for magnetic/flux excitations.

The symmetries of interest for us are: the global matter symmetry $P = \prod_v X_v$ and the magnetic 1-form symmetry $W_\gamma = \prod_{l \in \gamma} \sigma_l^z$, running parallel to bonds of the lattice. The latter is only a symmetry if $h = 0$ and it is generated by Wilson loops[22]. Indeed, we can equate $\sigma^z = e^{iA}$ with the gauge field (such that $W_\gamma$ is indeed a Wilson line), whereas $\sigma^x = e^{i\pi E}$ is the electric field [7].

We will follow the convention that the 1-form symmetry is denoted as $\mathbb{Z}_2[1]$; more generally, a $p$-form symmetry $G$ is denoted as $G[p]$, where the more familiar case of global symmetries is that of $G[0] = G$. While such higher-form symmetries have been well-studied [19–21, 61, 62][23], they might seem strange since in a sense they are infinitely fine-tuned (in that most perturbations would explicitly break them). The reason the notion still has merit is that unlike usual global symmetries, the consequences of higher-form symmetries can sometimes persist even when they are explicitly broken. Indeed, as mentioned in the introduction, the deconfined (or topologically ordered) phase can be characterized as spontaneously breaking the above magnetic 1-form symmetry, but its resulting bulk degeneracy (depending on the genus of space) is a robust feature of the phase, even when the symmetry is explicitly broken. In fact, in the low-energy field-theory description, the higher-form symmetry re-emerges. We will see that a similar feature pertains to the Higgs phase, i.e., the non-trivial SPT phase protected by the higher-form symmetry—its edges mode will display a certain robustness, even when the infinitely fine-tuned symmetry is explicitly broken.

## 4.2 SPT phase and string order parameter

Let us first consider the limit $J \to \infty$ and $h \to 0$. We obtain a ground state $|\psi\rangle$ which is characterized by two types of stabilizers, one for each vertex and the other for each link[24]:

$$
\begin{array}{cc}
\sigma^x \\
\sigma^x\,X\,\sigma^x = 1 & \text{and} & Z\,\sigma^z\,Z = 1. \\
\sigma^x
\end{array}
\tag{20}
$$

Of course the former is simply the Gauss law; the latter is the charge-hopping term[25].

These stabilizers are reminiscent of the 1D cluster chain we encountered in Sec. 3. In fact, these are exactly the two stabilizers defining a 2D SPT phase protected by a 0-form (here $P$) and 1-form (here $W$) symmetry! E.g., see Ref. [22] for the model on the triangular lattice instead of square lattice (which is in the same SPT phase). This suggests that we can think of the Gauss law an SPT stabilizer; this is the viewpoint that also arose in

---

[22]For contractible loops this is just a product $\prod_{p \in S} B_p$ such that $\gamma = \partial S$, but of course it is also a symmetry for non-contractible loops.

[23]There is a slight mismatch with the terminology: in the high-energy literature one typically only considers the non-contractible loop operators as being the symmetry generators; contractible ones are regarded as equal to the identity. In a lattice model, one might regard every loop operator as a symmetry generator. If one fixes all local loops to be in the +1 sector, one would recover the field-theoretic viewpoint.

[24]I.e., these are commuting operators, each squaring to the identity, and $|\psi\rangle$ is defined as the (unique) state which is left invariant under all of them.

[25]We also have the stabilizer $B_p = 1$, but this is not independent, as it can be written as a product of the link stabilizers.

Sec. 3.2.1. The main subtlety is that by definition of the gauge theory, this SPT stabilizer is *inviolable*, which ironically can make it easier to overlook its SPT nature.

By making $J$ finite, we tune away from the fixed-point limit, such that the second stabilizer in Eq. (20) no longer applies. However, as long as $h = 0$ (and until we hit a quantum phase transition), we retain long-range order in the SPT string order parameter:

$$\left\langle \quad Z \; \sigma^z \quad \sigma^z \quad \sigma^z \quad \sigma^z \quad \sigma^z \, Z \quad \right\rangle \neq 0. \tag{21}$$

Indeed, this is the gauge-invariant statement for saying that the gauge charge has condensed. From an SPT point of view, it can be interpreted as saying that magnetic 1-form symmetry is preserved, such that its domain wall operator has long-range order. Indeed, by usual symmetry fractionalization arguments [13, 44], one can argue that if $W$ is unbroken[26], the open Wilson string *must* have long-range order for some choice of endpoint operator[27]. In Sec. 4.4.1 we will see how the non-trivial $P$-charge of this endpoint operator implies edge modes.

## 4.3 Phase diagram with periodic boundary conditions: toric code in a field

Before exploring the consequences of the Higgs phase being an SPT phase, we first briefly recap the known phase diagram for periodic boundary conditions.

It is possible to disentangle matter and gauge field variables with a finite-depth unitary transformation (the net result is equivalent to choosing the so-called unitary gauge). In particular, we define $U = H \times \prod_v \prod_{l \in v} (CZ)_{l,v} \times H$, where $CZ$ is the "Controlled-$Z$" gate[28] and $H = \prod_l \frac{\sigma_l^x + \sigma_l^z}{\sqrt{2}}$ is the Hadamard transformation on each link. Then $U G_v U^\dagger = X_v$, so the Gauss law is now simply $X_v = 1$. This pins the matter fields, and the remaining link variables are now physical (i.e., unconstrained by any Gauss law) with Hamiltonian:

$$U H U^\dagger = -\sum_v A_v - \sum_p B_p - J \sum_l \sigma_l^z - h \sum_l \sigma_l^x. \tag{22}$$

(We remind the reader that $A_v$ and $B_p$ are defined in Eq. (17).)

We thus obtain the toric code model in a field. Its phase diagram has been explored [5, 45–47] and is reproduced in Fig. 1(a). In particular, we can easily see that the Higgs and confined regions are adiabatically connected [7], since if $J = \rho \cos\theta$ and $h = \rho \sin\theta$, then for $\rho \to \infty$, the ground state is a product state throughout: $|\psi\rangle = \otimes_l (\cos(\theta/2)|\uparrow\rangle_l + \sin(\theta/2)|\downarrow\rangle_l)$.

Note that this product state is not inconsistent with the Higgs phase being an SPT phase for $h = 0$. After all, SPT phases can be trivialized by finite-depth circuits. The defining property of SPT phases is that they cannot be trivialized by circuits which commute with the protecting symmetry. Note that the gates making up $U$ indeed do *not* commute with $P$ or $W$. One physical diagnostic of the fact that the Higgs phase is an SPT, is by probing its edge modes; indeed, with open boundaries we will find that it *cannot* be a product state.

---

[26]We say $W$ is spontaneously broken if we have long-range order in string operators which are charged under $W$. This implies topological order [20]. Hence, a state which preserves $W$ symmetry has short-range entanglement.

[27]The gist is as follows: if closed $W$ loops leave the state invariant and the wavefunction is short-range entangled with a finite correlation length $\xi$, then open $W$ strings of length $l \gg \xi$ can only affect the wavefunction near the endpoints of the string. I.e., it has the effective action $W_L W_R$ where, say, $W_L$ is localized near the left endpoint (of radial support $\xi$). Hence, $W_L^{-1} \times (W\text{-string}) \times W_R^{-1}$ leaves the state invariant.

[28]Since $(CZ)_{a,b} = (CZ)_{b,a}$, it does not matter whether one controls on the link or vertex qubit.

## 4.4 Edge modes

We now return to the original Fradkin-Shenker model in Eq. (19) and consider it with boundaries. It is simplest to take 'rough' edges since then the usual Gauss law is well-defined for every vertex (see Sec. 4.4.5 for a discussion of other cases). Let us thus consider the following half-infinite geometry:

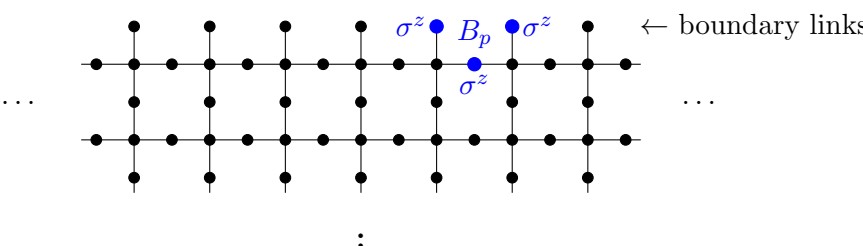

We take the same Hamiltonian as Eq. (19), where we interpret $B_p$ for the boundary plaquettes as a product over the *three* links, as shown. Crucially, note that the $J$-term does *not* appear for the boundary links, since those links are only adjacent to a *single* vertex[29]. This can also be called the 'electric' boundary condition, since Wilson lines can end at the boundary.

A key point is that in the presence of boundaries, $P$ is not purely gauge: multiplying all Gauss operators gives $1 = \prod_v G_v = \prod_{v \in \Lambda} X_v \times \prod_{l \in \partial \Lambda} \sigma_l^x$ where by $\Lambda$ we mean the lattice of all vertices, and $\partial \Lambda$ are all the links sticking out at the edge. Thus, by virtue of the Gauss law, $P$ is an honest global symmetry which however acts only on the boundary links:

$$P = \prod_{v \in \Lambda} X_v = \prod_{l \in \partial \Lambda} \sigma_l^x. \tag{23}$$

We note that the same Hamiltonian (including boundary terms) was considered by Harlow and Ooguri in Ref. [35]. Moreover, it was pointed out that the matter symmetry acts on the boundary links, as in Eq. (23), where it was interpreted as an asymptotic symmetry[30].

### 4.4.1 With magnetic symmetry ($h = 0$)

In the presence of boundaries there is a degeneracy due to the interplay of the $P$ and $W$ symmetries. We have already discussed that $P$ effectively acts on the boundary links as $\prod \sigma^x$ (see Eq. (23)). Note that this anticommutes with $W$ whenever they intersect:

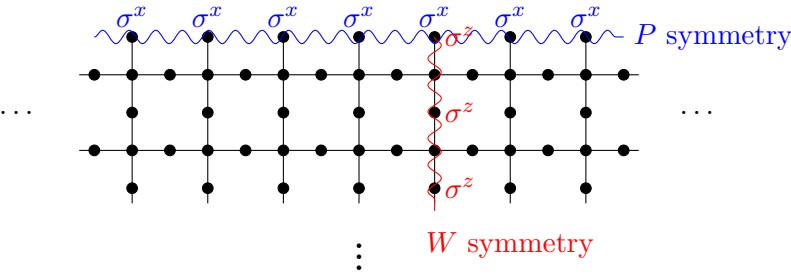

---

[29]We could include a term $-J\sigma_l^z Z_v$ for those links, which is indeed gauge-invariant, but it explicitly breaks the matter symmetry $P = \prod_v X_v$, which we want to respect.

[30]Footnote 54 of Ref. [35] even mentions in passing that this asymptotic symmetry is not respected in the ground state of the Higgs phase (in a particular limit), but the boundary phase diagram of the Fradkin-Shenker model was not explored.

Indeed, whenever one has two anticommuting symmetries, there is a degeneracy[31]. Note that the above $P$ and $W$ lines are symmetries as long as $h = 0$, so the above comment applies to both the Higgs (i.e., large-$J$) and deconfined (i.e., small-$J$) phases. Indeed, the deconfined phase is topologically ordered and has ground state degeneracy on the cylinder geometry, where $W$ in the above figure can end on the other boundary (not shown). However, for a rectangular geometry (with a single connected boundary), the $W$ line can only end on the same boundary, in which case it commutes with $P$; indeed, the deconfined (topological) phase has a unique ground state in this geometry. In contrast, in the Higgs phase, a $W$ line can terminate in the bulk (Eq. (21), or more precisely see footnote 27), since we have a charge condensate, leading necessarily to a twofold degeneracy. Moreover, since the bulk is short-range entangled, this degeneracy is located entirely on the edge (in contrast to the topological degeneracy of the deconfined phase). More rigorously, we can derive that the edge theory has long-range order for a local order parameter which is charged under $P$, as demonstrated in Fig. 6.

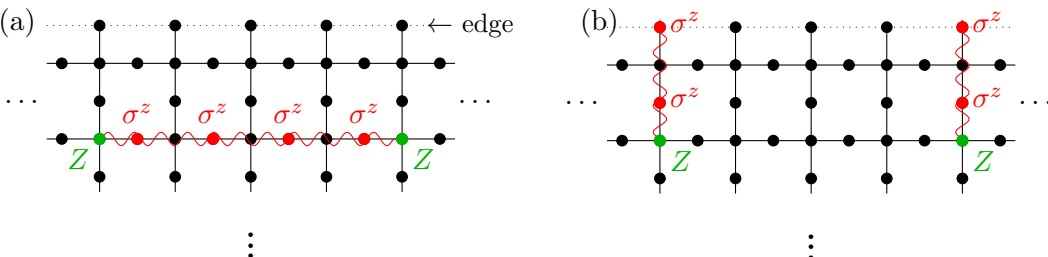

Figure 6: **Edge mode from the Higgs condensate order parameter.** We consider the Fradkin-Shenker model (19) on a half-infinite geometry with boundary links sticking out. We tune to the Higgs phase, in the special case that $h = 0$, such that we have exact magnetic symmetry. (a) In the bulk we are guaranteed[27] long-range order in the open Wilson string, since this is the bulk SPT order parameter (see Sec. 4.2). Since this is nonlocal, this does not imply any bulk degeneracy. (b) By using the fact that the system has magnetic 1-form symmetry, we can move the Wilson lines into the boundary (by multiplying with a symmetry generator). We now obtain long-range order for a *local* order parameter which is charged under the global matter symmetry $P$ (in the above case, it is an operator of the form '$\sigma^z\sigma^z Z$'). This implies a symmetry-breaking degeneracy along the boundary.

We note that there is also a slightly different perspective on why the presence of the magnetic one-form symmetry forbids restoring the symmetry of the ground state under $P = \prod_{l \in \partial \Lambda} \sigma_l^x$. Indeed, since any Wilson line operator that ends on the boundary must commute with the Hamiltonian, magnetic flux is static not only in the bulk, but also cannot be changed at the boundary. As a result, we cannot add terms which would condense boundary domain wall operators (strings of $\sigma^x$ acting on boundary links) since their endpoints would toggle flux. Hence, $W$ symmetry prevents us from restoring the symmetry of the ground state under $P$.

---

[31]Proof: Let the unitary operators $A$ and $B$ be anticommuting symmetries. Suppose $|\psi\rangle$ is an energy eigenstate. Note that $|\psi\rangle, A|\psi\rangle, B|\psi\rangle$ all have the same energy. Moreover, they cannot span a one-dimensional space, since otherwise $A|\psi\rangle = a|\psi\rangle$ and $B|\psi\rangle = b|\psi\rangle$, leading to a contradiction: $|\psi\rangle = aba^{-1}b^{-1}|\psi\rangle = aba^{-1}B^\dagger|\psi\rangle = B^\dagger aba^{-1}|\psi\rangle = B^\dagger A^\dagger BA|\psi\rangle = -|\psi\rangle$.

### 4.4.2 Without magnetic symmetry ($h \neq 0$): solvable line $J \to \infty$

Thus far, we have explored how in the presence of magnetic symmetry ($h = 0$), the Higgs phase is an SPT phase, which gives rise to edge modes. Here, we show that explicitly breaking $W$ does not immediately destroy the edge modes of the SPT phase. The deeper reason for this is that $W$ is a higher-form symmetry, and at low energies they are difficult to destroy, since by definition there are no local operators charged under them (since objects charged under a $p$-form symmetries are $p$-dimensional). Of course, on the lattice there is no crisp distinction between point-like operators or 'very small string operators', which is why we will explicitly verify the persistence of the edge modes in the lattice model.

Let us first consider a solvable case: we set $J \to \infty$ (in Eq. (19) with rough boundaries as discussed above) but keep $h$ arbitrary. This pins the bulk variables: only the boundary links remain, since $J$ is not present there (as matter cannot hop off the system due to $P$ symmetry). In particular, using the unitary[32] $U$ (see Sec. 4.3), we obtain the following Hamiltonian on the link variables (we have thrown out all terms which do not commute with $J$ since $J \to \infty$):

$$\lim_{J \to \infty} U \left( H_{\text{open b.c.}} \right) U^{\dagger} = -J \sum_{l \notin \partial \Lambda} \sigma_l^z - \sum_p B_p - h \sum_{l \in \partial \Lambda} \sigma_l^x \tag{24}$$

$$= -J \underbrace{\sum_{l \notin \partial \Lambda} \sigma_l^z}_{\text{bulk}} - \underbrace{\sum_{\langle l, l' \rangle \in \partial \Lambda} \sigma_l^z \sigma_{l'}^z - h \sum_{l \in \partial \Lambda} \sigma_l^x}_{\text{boundary}} . \tag{25}$$

In particular, a boundary plaquette $B_p$ reduce to an Ising coupling for the boundary links (since the third link variable is pinned by the bulk term). We thus see that the boundary Hamiltonian is an effective 1D quantum Ising chain in a transverse field. This implies that $P$ remains spontaneously broken at the edge for $|h| < 1$. At $|h| = 1$, the boundary undergoes a quantum critical point, described by an Ising conformal field theory with central charge $c = 1/2$. For $|h| > 1$, we obtain a unique ground state, which smoothly connects to the trivial confined regime.

### 4.4.3 Without magnetic symmetry ($h \neq 0$): numerical study

Thus far we have seen that there are edge modes for the region of the Higgs phase where there is exact magnetic symmetry, and for large $J$ we were able to analytically prove that these modes remain robust upon explicitly breaking magnetic symmetry (up until $h \approx 1$).

In fact, for any $J$ in the Higgs phase, it is clear that the edge modes cannot immediately gap out when turning on $h \neq 0$. After all, for $h = 0$ the boundary theory spontaneously breaks[33] $P$, and explicitly breaking $W$ cannot immediately undo this. As long as we preserve $P$, the boundary must undergo a gap-closing in order to achieve a trivial boundary theory. One can say that the boundary theory will have an emergent $W$ symmetry (generated by the local order parameter along the boundary) until there is a gap-closing; indeed, this is made precise by studying the boundary anomaly in Sec. 4.8.

This general argument shows that as long as we start with a $P$- and $W$-symmetric boundary Hamiltonian for $h = 0$, then there will be an *open region* in parameter space where the Higgs "phase" has edge modes. The *exact* location of the boundary transition

---

[32]Since we are discussing an SPT, it is important to *first* consider the Hamiltonian with open boundaries and to *then* use the unitary transformation; reversing these two steps can lead to the erroneous conclusion that the phase and boundary is trivial.

[33]Note that the (generalized) Mermin-Wagner-Elitzur theorem [63] forbids $W$ from spontaneously breaking at the edge.

(in parameter space) is not universal and depends, for instance, on the choice of boundary couplings. Here we determine it numerically for the above choice of boundary Hamiltonian.

To study this, we employ the infinite Density Matrix Renormalization Group (iDMRG) method [64], which can be adapted to study two-dimensional systems [65].[34] Since we are interested in the boundary theory, we consider an infinitely-long strip of width $L_y$, with two parallel "rough" boundaries. Using the TenPy library [66] we obtain the ground state of the Hamiltonian for arbitrary $J$ and $h$ and for different choices of $L_y = 1, 2, 3$. We find that already for the last two values the results do not differ significantly, so that we can be confident about the extrapolation to the 2d limit without having to push our numerics to computationally challenging system sizes. In order to probe the different phases, we consider a number of observables. To detect the edge transition, we measure the boundary order parameter $\langle \sigma^z_{\text{edge}} \rangle$, which is non-vanishing only in the Higgs phase. Another interesting observable that characterizes the system is the correlation function $\langle \sigma^z_U \sigma^z_L \rangle$ between spins on the upper and lower edges of the strip. While outside of the Higgs phase the individual expectation values $\langle \sigma^z_{\text{edge}} \rangle$ always vanish, this correlation function should only vanish in the confined phase. Finally, we point out that transitions which are characterized by a closing of the energy gap can be detected by looking for peaks in the entanglement entropy and correlation length $\xi$. Both these quantities are easily accessible when the ground state is written as a Matrix Product State (MPS), and allow to promptly identify boundaries between different gapped phases.

All the aforementioned observables confirm our predictions. While we present numerical details in Appendix A, the main resulting boundary phase diagram is shown in Fig. 1(b). The blue shaded regions denote where we find a nonzero order parameter (indeed, since the strip is infinitely-long, DMRG spontaneously breaks the symmetry rather than forming a cat state). The red line denotes a $1 + 1d$ Ising criticality. The phase boundaries of the deconfined (toric code) phase are also found to be in agreement with the well known results of Fradkin and Shenker [5, 7, 45–47]. Note however in contrast to the bulk phases and transitions, the boundary phase and transition in the shaded region has not, to the best of our knowledge, been commented on or reported before. Intriguingly, the boundary transition seems to merge with the enigmatic bulk multicritical point of the phase diagram [5, 45–47]; at this point in time we cannot conclusively establish that these indeed merge, and this remarkable feature deserves further study.

### 4.4.4 Breaking matter symmetry

As argued above, a mild breaking of the magnetic one-form symmetry does not lift the edge degeneracy. Physically, creating vison pairs at the boundary—which are *gapped* local magnetic excitations in the Higgs phase—cannot not appreciably split the energies of the two degenerate states. More precisely, the splitting is generated in perturbation theory at an order that is extensive in system size, leading to an exponentially small finite-size splitting which is parametrically suppressed by the vison gap.

Due to the presence of the Higgs condensate of matter in the bulk, breaking the global (i.e., 0-form) matter symmetry $P$ has no such robustness. Imagine breaking the matter symmetry by pushing a charge through the boundary. This is achieved by adding to the Hamiltonian a term that creates a Wilson line emanating from the boundary and ending with a matter charge (such as in Fig. 6(b)). Due to the Higgs matter condensate with its string order parameter, the vacuum expectation value of the string is finite and its sign distinguishes the two spontaneously broken states. Consequently, the energy splitting

---

[34]For this to be possible one dimension needs to be finite or compact, so that the system can be mapped to a 1d chain with long range interactions.

caused by this perturbation scales linearly in its strength and immediately opens a gap. This is customary for SPT phases protected by global symmetries: edge modes tend to immediately disappear when violating said symmetry.

### 4.4.5    Changing the boundary Hamiltonian

Thus far, we discussed a particular boundary Hamiltonian with a 'rough' edge (i.e., with gauge links sticking out). However, the 'Higgs=SPT' phenomena (and its associated edge modes) is more generally applicable; it only relies on choosing a boundary Hamiltonian which commutes with $P$ and $W$ symmetry. Indeed, the arguments given above already show that no matter what symmetry-preserving boundary perturbations we include for the rough edge, there will be an open region of the Fradkin-Shenker phase diagram with protected edge modes—delineated by a boundary phase transition.

What happens if we would choose, say, a smooth boundary? The results depend on whether we enforce the Gauss law on the boundary sites. Note that now there are only three gauge links emanating from a boundary site, in which case the Gauss operator no longer commutes with the magnetic symmetry. In this case we do not expect edge modes, since we have strongly violated the symmetries protecting the SPT phase.[35] If one does not enforce the Gauss law on the boundary, then we restore $W$ symmetry, thereby leading to protected edge modes (see Appendix B for a detailed analysis which confirms this prediction). This set-up can arise in a variety of ways, such as an effective description of an SIS interface in the bulk (Sec. 4.5) or of a spatial interface between distinct SPT phases (Sec. 4.6.3).

There is also a more general and universal perspective on why and when the edge mode emerges. One can interpret the $P$ symmetry action on the boundary as measuring the electric flux piercing the boundary (indeed, see Eq. (23)). Hence, requiring the boundary Hamiltonian to commute with $P$ and $W$ corresponds to the physical demand that electric and magnetic charges are globally conserved. These conservation laws carry a mutual anomaly at the edge (forbidding a trivial edge), which in turn can be traced back to the *bulk* anomaly of the deconfined phase of the gauge theory (this is discussed in detail in Secs. 4.5 and 4.8). This perspective generalizes to other gauge groups, as we discuss in Part II [27].

## 4.5    Bulk probe: localized zero-energy modes at an SIS junction

We have seen how the bulk SPT phase results in an edge mode, in line with the bulk-boundary correspondence. However, studying gauge theories with hard boundaries is a rather subtle endeavour, and it is thus desirable to also have a bulk probe. Here we discuss how this is possible by effectively generalizing the aforementioned bulk-boundary correspondence into a bulk-*defect* correspondence. This is quite a general (and novel) way of probing SPT phases, which we elaborate on in Sec. 5.1.

Let us recall that the Higgs phase is an SPT phase protected by the magnetic 1-form symmetry and the global matter symmetry. While the former is a gauge symmetry in the bulk, at the boundary there exist local charged operators, namely, short Wilson strings. We found that this physical matter symmetry was spontaneously broken at the edge. One way of making the matter symmetry physical in the bulk is by introducing a

---

[35]For a smooth boundary it is straightforward to construct an Ising gauge theory with dynamical vison matter. In such theory a smooth boundary does not break its 1-form electric symmetry. After condensing visons, we end up with an SPT order protected by the electric 1-form and the 0-form vison matter symmetries. This is in essence the $e - m$-dual of the scenario discussed above.

defect line across which matter is not allowed to tunnel. We call this an 'SPT-insulator-SPT', or 'SIS', junction. For concreteness, let us say this line defect is at $x = x_0$. Then $P_{\mathrm{SIS}} = \prod_{v \text{ with } v_x \leq x_0} X_v$ is a symmetry of the model[36]. Note that $P_L$ is *not* a gauge symmetry. E.g., any finite Wilson string (which is gauge-invariant) which has one endpoint on each side of the junction is charged under $P_L$. The SPT phase results in $P_L$ being spontaneously broken at this junction, implying a two-fold ground state degeneracy.

More precisely, the Hamiltonian with this SIS junction, $H_{\mathrm{SIS}}$, is simply given by Eq. (19) where we omit any term which does not commute with $P_L$. I.e., we throw out all the charge hopping terms $Z_v \sigma^z_{v,v'} Z_{v'}$ where $v$ and $v'$ are on opposite sides of the junction. To see that this leads to a degeneracy, let us first consider the fixed-point limit $J \to \infty$ and $h = 0$. In this case, the charge-hopping terms we have thrown away are stabilizers; their commutation with $H_{\mathrm{SIS}}$ and anti-commutation with $P_L$ implies a degeneracy. Upon making $J$ finite, we can no longer appeal to these ultra-short Wilson strings (as they no longer commute with $H_{\mathrm{SIS}}$). However, as long as we remain in the Higgs phase, a longer Wilson string (spanning across the defect line) will leave the ground state invariant[37], with appropriately dressed endpoint operators[38] (see Sec. 4.2), the left one being odd under $P_L$, so the degeneracy is stable. See Fig. 5 for the lower-dimensional analogue.

We thus see that an insulating defect line in the Higgs phase (with magnetic symmetry, i.e., $h = 0$) leads to the spontaneous breaking of the conserved charge $P_{\mathrm{SIS}}$, its order parameter being the open Wilson line straddling the junction. Similar to the case with the hard boundary, this is robust upon introducing magnetic excitations, i.e., $h \neq 0$. Indeed, the explicit breaking of magnetic symmetry cannot immediately restore $P_{\mathrm{SIS}}$; one has to drive a defect phase transition where the magnetic fluctuations disorder the Wilson line crossing the junction, which we can think of as a sort of confinement on the wall. This results in a phase diagram similar to the boundary phase diagram in Fig. 1. In fact, one can make this connection more precise: if we drive the region on the right-hand side of the defect into the fixed-point limit of the Higgs phase (i.e., $J \to \infty$ and $h \to 0$), then the remaining $H_{\mathrm{SIS}}$ coincides with the boundary Hamiltonian studied in Sec. 4.4!

This shows that the exotic features which we usually associate to the boundaries of an SPT phase can also emerge at a defect line in the bulk. While such an SIS defect is quite novel from the SPT perspective, it is of course quite standard for Higgs phases. In particular, in addition to supercurrents and the Meissner effect, superconductors are famous for exhibiting the Josephson effect [67] upon introducing an insulating region within the superconductor. In Part II [27], we interpret these aspects of $U(1)$ gauge theory as an SPT phenomenon. Conversely, in Sec. 5.1 of the present work, we show how general SPT phases can be probed in the bulk via such an SIS junction.

As an aside, it is intriguing to note that SIS junctions can be used to reveal the topological properties of the Higgs phase *even without an explicit matter symmetry*, since the junction itself comes naturally equipped with a symmetry (e.g., see Fig. 3). Recall that the magnetic symmetry arises from the absence of magnetic monopoles or other dynamical magnetic excitations. Inside the ideal SIS junction, across which no gauge charge is allowed to tunnel, there are also no electric excitations, and we get a corresponding symmetry called the electric or center symmetry. Intuitively, this symmetry measures the electric flux across the SIS junction. Because of the commutation relation between electric and magnetic fields, this symmetry shares an anomaly with the magnetic symmetry, which protects special modes on the junction. For a large insulating junction, we can identify

---

[36]Note that by gauge symmetry, it does not matter whether we take the left or right half.

[37]Here we are only using that the ground state respects the 1-form symmetry; in particular, it is not spontaneously broken, thereby excluding the deconfined phase.

[38]These endpoint operators are dressed versions of $Z_v$; see Sec. 3.2.3 for explicit expressions in the 1D case.

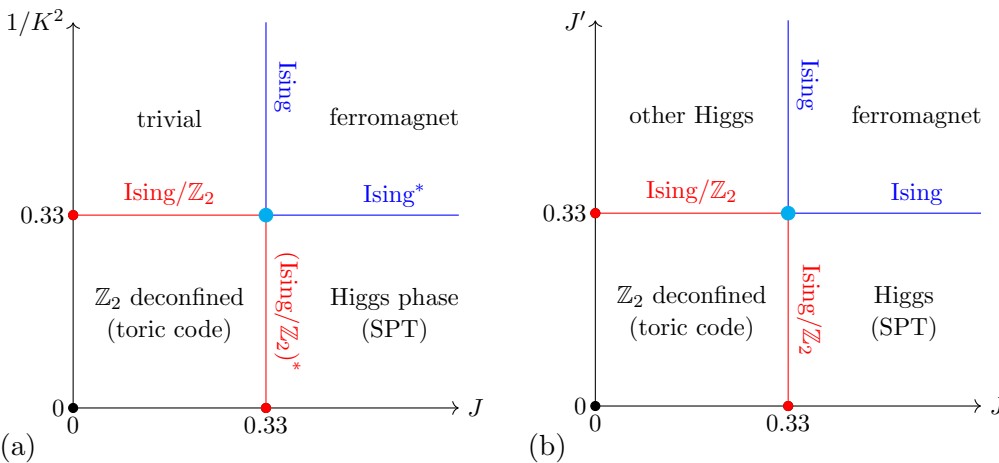

Figure 7: **Phase diagrams with distinct SPT phases.** (a) Emergent $\mathbb{Z}_2$ gauge theory (26) where a Gauss law emerges energetically for $K \to \infty$. For small $K$, we can access a product state outside of the gauge theory Hilbert space. This is separated from the Higgs phase (which is a non-trivial SPT phase) by a bulk phase transition. For this particular model, the cyan dot is a CFT described by Ising$^2/\mathbb{Z}_2$ (orbifold by diagonal Ising symmetry). (b) Phase diagram for a gauge theory (i.e., Gauss law is valid everywhere) with *two* matter fields coupled to the same $\mathbb{Z}_2$ gauge field (28). Their respective Higgs condensates are *distinct* SPT phases. While they have unique ground states (in the absence of boundaries), a spatial interface between the two Higgs condensates will have ground state degeneracy.

these with the deconfined gauge field inside the junction, see Sec. 4.8. In fact, this can be made more explicit by giving the insulating region a larger width $w$, in which case we recognize it as a sliver of the deconfined ('toric code') phase. From this perspective, we can interpret the SPT edge modes as the topological degeneracy of the deconfined phase. For generic perturbations, this degeneracy will be split exponentially in $w$ by an $e$-anyon crossing the junction, i.e., a Wilson line stretching from one side to the other. $P_{\text{SIS}}$ forbids precisely this process, which allows the degeneracy to persist even as the width $w$ goes to 0.

## 4.6   Bulk SPT phases: transitions and interfaces

Thus far we have focused on lower-dimensional probes of the fact that the Higgs phase is a non-trivial SPT phase—i.e., symmetry-breaking degeneracies at the boundary or at bulk defects. However, usually we also associate a non-trivial SPT phase to there being a bulk critical point separating it from a trivial (or other) SPT phase. Here the subtlety of gauge theory comes in: in the minimal model under consideration (a gauge theory with one Higgs field), there can be only *one* short-range entangled state with $P$ and $W$ symmetries. This traces back to the inviolable Gauss law: whenever $W$ symmetry is unbroken, its open string must have long-range order, but by the Gauss law, its endpoint must be non-trivially charged under $P$, which specifies the SPT class. This viewpoint is discussed in more detail in the lower-dimensional analog in Sec. 3.

    Naturally, if there is no other SPT phase to compare it to, there is also no bulk critical point. One option (as in the above Fradkin-Shenker model) is to compare the Higgs phase with the confined phase. But the latter does not respect $W$ symmetry, i.e., tuning $h \neq 0$

trivializes the SPT class of the Higgs phase, which is the very reason that the Higgs and confined phases are indeed adiabatically connected for periodic boundary conditions [7]. While we found that various SPT properties are still parametrically robust, there is no *bulk* phase transition to explore in that setting.

There are two natural modifications which do allow for *another* SPT phase whilst preserving $P$ and $W$, which we explore here. The first is to simply make the Gauss law emergent, such that there exists states in the Hilbert space which do not respect it. The second route is to keep the Gauss law exact, but introduce a second matter field. In both cases, we will find a bulk SPT phase transition, which cannot be avoided as long as $P$ and $W$ are preserved (indeed in the following two subsections we set $h = 0$, i.e., we consider exact magnetic symmetry). Moreover, we can then re-interpret the edge modes of the Higgs phase as arising at the spatial interface between these distinct SPT phases.

### 4.6.1   Emergent gauge theory

Suppose the Gauss law is not an intrinsic property of the Hilbert space, but rather arises energetically from an additional term in Eq. (19):

$$H = -\sum_v X_v - \sum_p B_p - J \sum_{\langle v,v'\rangle} Z_v \sigma^z_{v,v'} Z_{v'} - K \sum_v G_v - \frac{1}{K} \sum_l \sigma^z_l. \tag{26}$$

The two limits of $K$ give us valuable insight into the above model. First, if we take $K \to \infty$, we energetically pin $G_v = 1$ and the model becomes indistinguishable from an 'actual' gauge theory (as for Bob in the thought experiment of Sec. 3). From our above analysis, we thus know that the (deconfined) toric code arises for small $J$ and the Higgs SPT phase for large $J$. In contrast, the other limit $K \to 0$ is very far away from an effective gauge theory, instead pinning $\sigma^z_l = 1$. In that limit, only the matter degrees of freedom remain which are described by the Ising model: $\lim_{K \to 0} H = -\sum_v X_v - J \sum_{v,v'} Z_v Z_{v'}$, with a product state for small $J$ (which is symmetric under both $P$ and $W$) and a phase which spontaneously breaks $P$ for large $J$.

It is worth noting that we can interpret the large-$K$ limit as the gauged version of the small-$K$ limit: indeed, gauging the Ising model (i.e., $\lim_{K \to 0} H$) leads exactly to $\lim_{K \to \infty} H$. The above model (26) thus effectively interpolates between a model and its gauged counterpart. In Ref. [33] this was dubbed 'gentle gauging' (where it was explored in the one-dimensional setting). Such an interpolation allows one to compare the phases. In particular, our SPT perspective implies the trivial phase (which arises for small $K$) should be separated from the Higgs phase (which arises at large $K$) by a quantum phase transition, despite both phases being short-range entangled. Indeed, the former has long-range order for a $W$ string whose endpoint is even under $P$, in contrast to what we saw in Sec. 4.2 for the Higgs phase.

To confirm this prediction, we obtain the phase diagram for Eq. (26). It turns out that this can be easily obtained by a Kramers-Wannier transformation of the link variables[39]. This leads to Fig. 7(a). We indeed find that the two short-range entangled phases are *not* adiabatically connected. In this particular model, we find they are separated by a direct (multi)critical point described by a $\mathbb{Z}_2$ orbifold of two copies of the 2+1D Ising universality class[40]. This splits apart into four distinct Ising critical lines; more precisely, two of these

---

[39]Introduce Pauli operators via the following nonlocal re-identification: $X_{A,v} = X_v$, $X_{B,v} = G_v$, $Z_{A,v} Z_{A,v'} = Z_v \sigma^z_{v,v'} Z_{v'}$, $Z_{B,v} Z_{B,v'} = \sigma^z_{v,v'}$. In these variables, we have two decoupled Ising models [68] with coupling constants $J_A = J$ and $J_B = 1/K^2$ for Eq. (26). Similarly, for Eq. (28) we obtain $J_A = J$ and $J_B = J'$.

[40]However, generic perturbations will cause a flow to $O(2)/\mathbb{Z}_2$ criticality [69, 70].

are the conventional 2+1D Ising criticality, whereas the other two are the gauged version. Moreover, the ones neighboring the Higgs phase have non-trivial symmetry-enrichment due to $P$ and $W$ symmetry, which will protect edge modes even at criticality [60].

### 4.6.2    Multiple Higgs fields

A similar situation arises by remaining in a gauge theory, but by introducing an extra matter field. This can be done in two equivalent ways: either we add an additional field which carries no gauge charge (such that the global physical symmetry is the product of $P$ and the parity of the new field), or we add an extra Higgs field with gauge charge (these two are related by an on-site basis-transformation). We take the latter approach: each vertex now has $X_v, Y_v, Z_v$ and $X'_v, Y'_v, Z'_v$ Pauli operators; the new Gauss law is:

$$\tilde{G}_v = X_v X'_v A_v = 1. \tag{27}$$

We can then consider a Hamiltonian which involves both these matter fields:

$$H = -\sum_v X_v - \sum_v X'_v - \sum_p B_p - J \sum_{\langle v,v' \rangle} Z_v \sigma^z_{v,v'} Z_{v'} - J' \sum_{\langle v,v' \rangle} Z'_v \sigma^z_{v,v'} Z'_{v'}. \tag{28}$$

Now $P = \prod_v X_v$ is a *physical* symmetry, even in the bulk of the system. One can interpret it as the *relative* matter parity; note that by the Gauss law $P' = \prod_v X'_v = P$ in the bulk. The Higgs phases for large-$J$ and large-$J'$ define distinct SPT phases in the presence of $W$ symmetry. Indeed, the $W$-string has long-rang order for different endpoints operators, whose charge for $P$ differs for the two phases.

     The phase diagram is obtained similarly as for the emergent case, and the result is shown in Fig. 7(b). We indeed observe a phase transition separating the two Higgs phases, confirming that they are distinct SPT phases protected by $P$ and $W$. The similarity between the two panels in Fig. 7 is not a coincidence: using $\tilde{G}_v = 1$, one can gauge-fix the second matter field away, $Z'_v = 1$. The result is that Eq. (28) then resembles an emergent gauge theory as in Eq. (26).

### 4.6.3    Edge modes at spatial interface between distinct bulk SPT phases

In Sec. 4.4, we studied the Fradkin-Shenker model with open boundaries and found that if the Hamiltonian respects the $W$ and $P$ symmetries, then the latter is spontaneously broken at the edge. Here we will show that the same 'boundary Hamiltonian' can in fact arise as the effective description of a system which has no boundary at all but rather a spatial interface between two distinct bulk phases. This gives us yet another confirmation of the non-triviality of the Higgs phase, without having to define a gauge theory with a hard boundary.

     In Sec. 4.6.1 and Sec. 4.6.2, we saw how making the Gauss law emergent or including a second Higgs field, respectively, gives two distinct bulk SPT phases. In fact, the two Hamiltonians were shown to be unitarily equivalent. Hence, without loss of generality, let us consider the Hamiltonian with an emergent gauge theory, Eq. (26). While the limit $K \to \infty$ effectively looks like the Fradkin-Shenker model, in the limit $K \to 0$ we have a distinct trivial SPT phase.

     Let us now consider the spatial interface between the two as shown in Fig. 8. In the limit $K, J \to 0$, we can replace $\sigma^z = 1$ and $X_v = 1$, shown in red. The remaining quantum system thus looks like the square lattice with a rough boundary. Moreover, the four-body plaquette term reduces to an effective three-body term, as shown. The resulting Hamiltonian is the one we studied in Sec. 4.4. In particular, all its properties carry over to this case. For instance, we learn that *a spatial interface in an exact gauge theory with two distinct Higgs fields carries an SPT edge mode!*

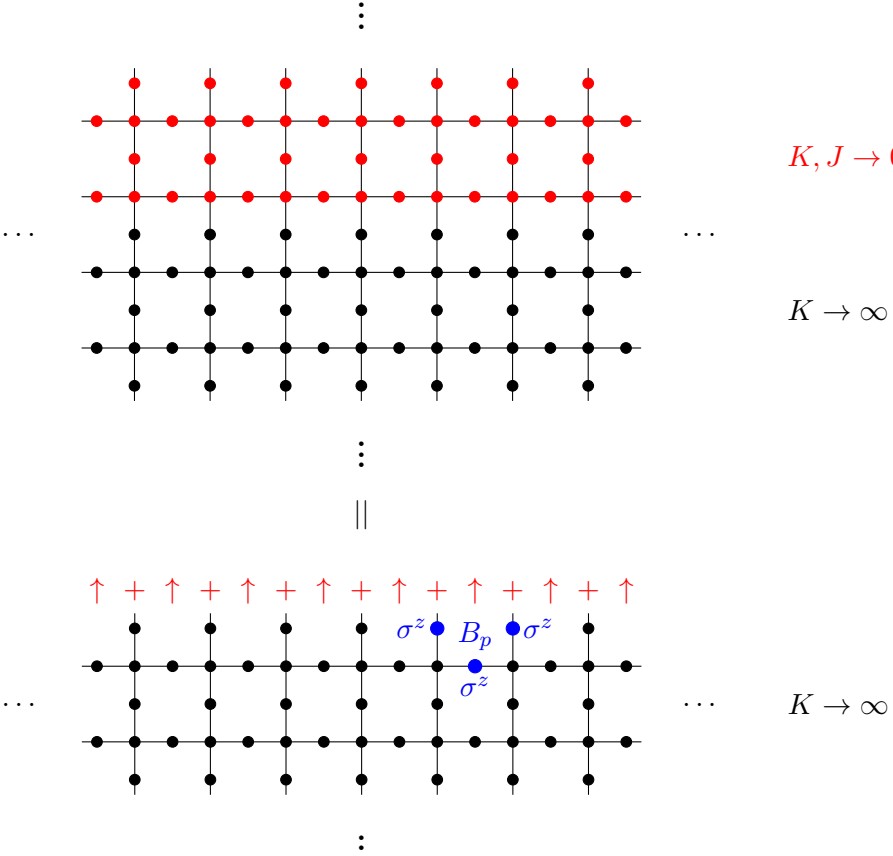

Figure 8: **Boundary Hamiltonian as a spatial interface.** If we have a gauge theory with an emergent Gauss law (Sec. 4.6.1), *or* two Higgs fields (Sec. 4.6.2), then there are two distinct bulk SPT phases. The spatial interface between the two carries a localized edge mode (Sec. 4.6.3). By driving one of the two regions into its fixed-point limit, the interface is effectively described by a system with a boundary and a particular boundary condition. This is one physical scenario which gives rise to the Fradkin-Shenker model with a (rough) boundary (Sec. 4.4).

## 4.7   Higher-dimensional Fradkin-Shenker models

The above discussion naturally generalizes to higher dimensions. For instance, repeating the same for $\mathbb{Z}_2$ lattice gauge theory in 3+1D, we first find that for $h = 0$ we have exact magnetic symmetry (still generated by Wilson loops). The gauge-invariant open Wilson line has long-range order in the Higgs phase. Since the endpoint of this SPT string order is charged under $P$, we are guaranteed edge modes for boundary conditions which preserve $P$ and $W$ (the rough boundary again being the natural one). Note that this edge still spontaneously breaks $P$, since the generalized Mermin-Wagner-Elitzur theorem forbids spontaneous breaking of $W$ [63] at the boundary. Moreover, similar to above, one can analytically show that in the $J \to \infty$ limit, turning on $h \neq 0$ still gives rise to robust symmetry-breaking along the edge—up to $|J| \approx 0.328$ where there is a 2+1D Ising critical point [68] associated to the 2+1D edge. Also for finite $J$, we again expect an open region of parameter space which supports edge modes (or interface modes at an SIS junction, or between the distinct SPT phases, as above), since the spontaneous breaking of $P$ is robust to explicitly breaking $W$.

## 4.8   Anomalous boundary action of symmetries in the $\mathbb{Z}_n$ Higgs phase

In this subsection, we offer a slightly more general perspective on why the Higgs phase has a protected edge mode. We show how the symmetry has an effective 'anomalous' action on the boundary, which stabilizes a symmetry-breaking edge in all dimensions. This perspective ties directly into the framework which is used to classify SPT phases [71–75], and it also readily extends to other gauge groups as we discuss in Part II for $U(1)$ and non-Abelian gauge theory [27].

A symmetry acting on a Hilbert space is said to be anomalous if its action is, in some sense, non-local. For example, if we have an SPT phase with a boundary, we can drive the bulk state into a fixed-point limit to obtain an effective Hilbert space for the edge—but the symmetry action will no longer be on-site [51, 71, 72, 75, 76]. The bulk SPT phase can be reconstructed from properties of these edge symmetry operators. This bulk-boundary correspondence, or anomaly in-flow, allows us to associate an SPT phase to a symmetry action and thereby speak of the anomaly of the symmetry as the equivalence class of this SPT[41].

A basic consequence of such an anomaly is that there are no symmetric short-range entangled states. One way of seeing this is by the lattice description of anomalies: there are certain discrete invariants associated to the circuit repesentations of non-onsite symmetries [72], whereas it is straightforward to show that the invariants are trivial if there exists a symmetric trivial state. This gives a strong constraint on the phase diagram of theories with an anomaly. At any value of the parameters we must have some combination of spontaneous symmetry breaking, gapless modes, or topological order—anything but short-range entanglement.

For example, if we study a 1d cluster chain (as in Eq. (1)) with open boundaries, and use the stabilizers to project out the bulk, we obtain a boundary symmetry action where the $\mathbb{Z}_2 \times \mathbb{Z}_2$ symmetry generators anti-commute. The projectiveness of the representation is not deformable, so the boundary degeneracy is robust. This symmetry action cannot be made linear (i.e., not projective) without violating locality, involving the other end of the cluster chain.

---

[41]There are attempts at more precise definitions. For instance, one can also say that the symmetry is anomalous (in the sense of 't Hooft) if it cannot be coupled to a background gauge field (but it can be gauged at the boundary of some SPT). This implies the symmetry is not realizable as an on-site symmetry, and the converse is thought to hold as well.

(a) 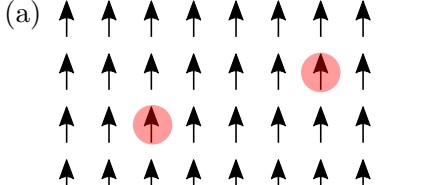     (b) 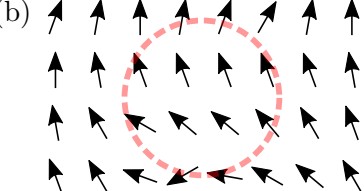

Figure 9: **Emergent higher-form symmetries in symmetry-breaking phases.** (a) Spontaneously breaking $\mathbb{Z}_n$ symmetry leads to a $d$-form symmetry (in $d$ spatial dimensions) which corresponds to the absence of domain walls. The order parameter $\mathcal{Z}_n$ has a stable value, e.g., its value is the same for the two red dots. (b) Spontaneously breaking $U(1)$ symmetry does not lead to a rigid order parameter since there are low-energy Goldstone modes which continuously deform the long-range order. However, there is an emergent $(d-1)$-form symmetry which characterizes the absence of vortices—which is indeed a quantized defect and hence absent in the ground state. In both cases, this emergent higher-form symmetry shares a mutual anomaly with the global symmetry; these are precisely the anomalies that emerge at the edge of the Higgs SPT phase (see Table 1).

We can draw an analogy between the projective action at the edge of the cluster chain and the 2d toric code, $H = -\sum_v A_v - \sum_p B_p$ (see Eq. (17)), which itself can live on the boundary of a 3d SPT. It has two 1-form symmetries given by $\prod \sigma_l^z$ on loops along the square lattice and $\prod \sigma_l^x$ on loops on the dual lattice (this latter symmetry is called the electric or center 1-form symmetry). The two symmetry generators anti-commute at each intersection, which means we get a projective symmetry action on the torus and higher genus surfaces. This anomaly is characteristic the edge of a 1-form $\mathbb{Z}_2[1] \times \mathbb{Z}_2[1]$ "cluster" SPT in 3d[42]. A consequence of the anomaly, as long as one preserves both of these 1-form symmetries, one cannot condense anyons and leave the long-range entangled phase.[43]

The boundary of the $\mathbb{Z}_n$ Higgs phase in $d$ space dimensions exhibits a third anomaly, for $\mathbb{Z}_n \times \mathbb{Z}_n[d-1]$, which is intermediate between these two, and has a very similar flavor in that the two generators anti-commute. (For $d = 1$ we have already argued in Sec. 3 that the Higgs phase reduces to the cluster chain.) This anomaly is characteristic of a $\mathbb{Z}_n$ spontaneous symmetry breaking state, where the $\mathbb{Z}_n[d-1]$ generator may be interpreted as an order parameter for the (0-form) $\mathbb{Z}_n$ generator.

We illustrate this with an example. Let $\mathcal{X}$ and $\mathcal{Z}$ denote the two clock operators for $\mathbb{Z}_n$ (for $n = 2$ these reduce to the Pauli operators). A $\mathbb{Z}_n$-symmetric Hamiltonian for spontaneous symmetry-breaking in $d'$ spatial dimensions is $H = -\sum_{\langle i,j \rangle} \mathcal{Z}_i^\dagger \mathcal{Z}_j + h.c..$ The global $\mathbb{Z}_n$ symmetry is $P = \prod_i \mathcal{X}_i$. However, note that this Hamiltonian also has a local $\mathbb{Z}_n[d']$ symmetry generated by $\mathcal{Z}_i$ (cf. the patch operators in Ref. [59].) The anti-commutation of $\mathcal{Z}_i$ and $\prod_i \mathcal{X}_i$ (i.e., the fact that the order parameter is charged) encodes a $\mathbb{Z}_n \times \mathbb{Z}_n[d']$ anomaly. We cannot trivially gap this system while preserving these symmetries.

This anomaly is more robust than it seems. Adding a transverse field $\mathcal{X}_i + \mathcal{X}_i^\dagger$ explic-

---

[42]We can sketch a model for this SPT on two interlinking cubic lattices $A$ and $B$ in 3d, with a qubit per edge. The stabilizers are of the form $Z$ on an $A$-qubit times a product of $X$ on the the square of $B$-qubits surrounding this edge, and vice versa switching $A$ and $B$, $X$ and $Z$. The symmetries are $\prod Z$ over $A$ lattice surfaces and $B$ links sticking out, and $\prod X$ over $B$ lattice surfaces with $A$ links sticking out, see also Ref. [22].

[43]This anomaly is actually a feature of many pure gauge theories. For example we see an analogous anomaly between the electric and magnetic flux conservation in pure $U(1)$ gauge theory, see Part II [27].

| | symmetry-breaking of $G$ = mutual anomaly in bulk | | Higgs phase for $G$ (= SPT) = mutual anomaly on edge | |
|---|---|---|---|---|
| $G = \mathbb{Z}_n$ | global symmetry: $\mathbb{Z}_n$ order parameter: $\mathbb{Z}_n[d']$ | | matter symmetry: $\mathbb{Z}_n$ Wilson line: $\mathbb{Z}_n[d-1]$ | |
| $G = U(1)$ | global symmetry: $U(1)$ winding: $U(1)[d'-1]$ | | matter symmetry: $U(1)$ magnetic flux: $U(1)[d-2]$ | |

Table 1: **Symmetry-breaking vs 'gauge symmetry-breaking' (i.e., Higgs phase).** Symmetry breaking phases have an emergent anomaly between the (broken) symmetry and either the order parameter (in the discrete case) or the winding number symmetry (in the $U(1)$ case); see Fig. 9. Although the Higgs phase is not a symmetry breaking phase of matter, it is an SPT phase whose boundary exhibits the aforementioned anomaly (with $d' = d_{\text{edge}} = d-1$ being the spatial dimension of the boundary), where the matter symmetry is spontaneously broken and the magnetic symmetry gives rise to the dual symmetry of the order parameter or its winding, respectively. That is, a Wilson line ends on a boundary order parameter, and magnetic flux induces a boundary winding. Note that if the bulk has magnetic symmetry, then the anomalous symmetry is exact, whereas if the former is broken, the latter is still emergent until we drive a (boundary) transition, as we saw in the Fradkin-Shenker phase diagram in Fig. 1. Note the magnetic symmetry is always at the critical dimension of the generalized Mermin-Wagner-Elitzur theorem on the boundary, and is thus never spontaneously broken [63].

itly breaks the $\mathbb{Z}_n[d']$ symmetry. However, as long as we remain in the symmetry-breaking phase, there will be an emergent $d'$-form symmetry; some exponentially-localized decoration of $\mathcal{Z}_i$ will still commute with the Hamiltonian (see Fig. 9(a)), up until the phase transition where the localization length (the correlation length) diverges. This is yet another exemplar of the stability of emergent higher-form symmetries.

This anomaly is realized at the edge of the Higgs phase, where $P$ is the global 0-form symmetry generator, and the $d'$-form symmetry generator is a Wilson line meeting the boundary with dimension $d' = d_{\text{edge}} = d-1$. When we project out the bulk, this Wilson line gets truncated as in Fig. 6(b). The resulting local symmetry generator carries a $P$ charge, so the two anti-commute as claimed. See the first row of Table 1.

Group cohomology is the mathematical tool for talking about such projective symmetry representations [77]. A mild generalization of group cohomology also lets us describe projective representations of higher form symmetries as above [21]. We can deduce the class of the boundary anomaly of the $\mathbb{Z}_n$ Higgs phase is

$$\Omega(A_{\text{mat}}, B_{\text{mag}}) = \frac{1}{n} A_{\text{mat}} \cup B_{\text{mag}} \in H^{d+1}(\mathbb{Z}_n \times \mathbb{Z}_n[d-1], U(1)), \tag{29}$$

where $A_{\text{mat}}$ generates $H^1(\mathbb{Z}_n, \mathbb{Z}_n)$, $B_{\text{mag}}$ generates $H^d(\mathbb{Z}_n[d-1], \mathbb{Z}_n)$, $\cup$ is the cup product of cohomology classes (analogous to a wedge product of forms), and this class should be understood as taking values in $\mathbb{R}/\mathbb{Z}$. When $d = 1$ this reduces to the class of the familiar projective representation of $\mathbb{Z}_n \times \mathbb{Z}_n$, matching the cluster chain in Section 3. $A_{\text{mat}}$ and $B_{\text{mag}}$ can be interpreted as gauge fields for the $\mathbb{Z}_n$ matter and $\mathbb{Z}_n[d-1]$ magnetic symmetries, respectively, and $2\pi\Omega$ as a topological term in the effective action of the Higgs-SPT bulk. We will derive it that way using field theory in Part II [27]. For now, we can note that the form of $\Omega$ directly encodes the charge at the end of the open Wilson line.

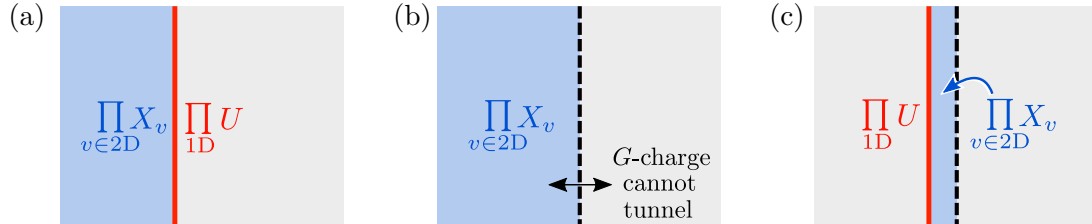

Figure 10: **Bulk-defect correspondence for an SPT phase.** Consider an SPT phase (in arbitrary spatial dimension $d$, where $d = 2$ in this figure) which is partially protected by a global symmetry $G$, e.g., a $\mathbb{Z}_2$ symmetry $\prod_v X_v$. (a) Since the state preserves the symmetry, the membrane operator $\prod X$ (blue) must have a nonzero expectation value by some appropriate dressing $\prod U$ along its boundary (red). By virtue of it being a non-trivial SPT phase, this $(d-1)$-dimensional operator $\prod U$ shares an anomaly with the (total) symmetry protecting the SPT phase. (b) We introduce a $(d-1)$-dimensional defect hypersurface (dashed) across which $G$ charge is not allowed to move. By definition, this means that, say, $\prod X$ to the left of the defect (blue) is a conserved symmetry. (c) By using the membrane order parameter mentioned in (a), the defect symmetry in (b) can be rewritten as a $(d-1)$-dimensional symmetry which is localized near the defect hypersurface. We observe that this is an anomalous symmetry, leading to localized edge modes.

Indeed, the background gauge field $B_{\mathrm{mag}}$ represents an insertion of closed Wilson lines in a spacetime $X$, via Poincare duality $H^d(X, \mathbb{Z}_n) = H_1(X, \mathbb{Z}_n)$. If we have an open Wilson line, then $dB_{\mathrm{mag}} \neq 0$ and $\Omega$ is not gauge invariant under $A_{\mathrm{mat}}$ gauge transformations unless we cancel it with the correct matter charge. This charge then relates to the boundary anomaly by studying gauge transformations of $\Omega$ on a manifold with boundary [21, 72].

# 5 Exporting insights which are applicable to general SPT phases

By re-interpreting the Higgs condensate as an SPT phase, we uncovered various phenomena which can be useful in the study of SPT phases—even those which are not directly related to a gauge theory. Here, we briefly summarize some of these key findings.

## 5.1 Generalized bulk-defect correspondence for SPT phases

The first insight which we want to generalize is that we were able to probe the SPT '*edge* mode' in the *bulk* by introducing a defect line across which charge was not allowed to tunnel (e.g., see Sec. 3.2.3 or Sec. 4.5). For simplicity, we phrase it for conventional global 0-form symmetries, but we will also discuss generalizations:

**Bulk-defect correspondence.** Consider a non-trivial SPT phase in $d$ spatial dimensions, which is (at least partially[44]) protected by a 0-form symmetry $G$. If we introduce a $(d-1)$-dimensional defect in the bulk across which $G$-charge is not allowed to move (Fig. 10), then this defect has an anomalous mode (i.e., it is symmetry-breaking, gapless, or topologically ordered).

For definiteness, let us say the total symmetry group protecting the SPT phase is $G \times H$. The simplest and most direct way of arguing the above bulk-defect correspondence is by reducing it to the usual case of an SPT edge mode. In particular, the $(d-1)$-dimensional defect mentioned above essentially implies that rather than just having $G$ symmetry, we have $G_L \times G_R$ symmetry. For instance, if the defect occurs at the hypersurface $x_1 = 0$, then $G_L$ is the $G$ symmetry acting on the left-hand side space $x_1 < 0$. Clearly, since the SPT phase is non-trivial for $G \times H$ symmetry, we learn that the left-hand side is also a non-trivial SPT phase for $G_L \times H$. On the other hand, the right-hand side $x_1 > 0$ must be trivial[45] for $G_L \times H$, since $G_L$ acts trivially on this part of space. In conclusion, both halves are in distinct SPT phases, and it is well-known that such spatial interfaces between distinct bulk phases lead to anomalous edge states.

Let us also present an alternative derivation of the anomalous state on the defect. This has the benefit of applying to more general defects (such as those associated to higher symmetries), although we will first discuss it for the 0-form case. The idea is summarized in Fig. 10. In panel (a) we remind the reader of the following two facts: (i) since our state is short-range entangled and symmetric, acting with $G$ on semi-infinite (hyper)membrane of $d$ spatial dimensions (shown in blue) has long-range order if we dress its $(d-1)$-dimensional boundary with an appropriate operator (shown in red) [44,76]; (ii) since the non-trivial SPT phase is (partially) protected by $G$, this boundary operator will have an anomalous action (upon taking account of the full symmetry group protecting the SPT phase) [72]. Upon inserting a no-tunneling defect for $G$-charges, we obtain that acting with $G$ on only a part of space is itself a symmetry (see Fig. 10(b)). Since the SPT phase has a finite correlation length $\xi$, then up to some finite width strip of order $\xi$ we can replace the semi-infinite hypermembrane by the aforementioned $(d-1)$-dimensional operator (Fig. 10(c)). In conclusion, we learn that the $(d-1)$-dimensional defect has an anomalous symmetry action, as claimed.

Observe that the above argument can be repeated for any $p$-form symmetry $G$, leading to a $(d-1-p)$-dimensional defect in space. In this higher case, extra care has to be taken to conclude whether and when the additional symmetry (associated to the defect) carries a non-trivial anomaly. In Appendix C we write down the anomaly class for the defect; in particular, from this one can deduce that in the case where the SPT response function is linear in the external gauge field for $G$, this anomaly class is nontrivial.

## 5.2 The surprising stability of higher-form SPT phases

The second insight which we want to generalize is that, in some sense, SPT phases protected by higher-form symmetries are parametrically robust to explicitly breaking said symmetry. We already saw an example of this in Fig. 1(b), where $h \neq 0$ explicitly breaks the higher magnetic symmetry but there is nevertheless an open region of parameter space with a non-trivial (degenerate) edge mode (see Sec. 4.4).

A more well-known statement is that in a phase which spontaneously breaks a higher symmetry, subsequently explicitly breaking said symmetry does not immediately destroy its degeneracy. Indeed, this is nicely explained in a recent review [20] which also touches on the topic of SPT phases protected by unbroken higher symmetries. However, the combination of these two—namely, SPT phases protected by higher symmetries which are

---

[44]By this we mean that $G$ alone need not be sufficient to protect a non-trivial SPT, but destroying it would change its SPT class.

[45]More precisely, the right-hand side can still be non-trivial for $G_L \times H$ if $H$ itself partially protects the SPT phase. What really matters is that we declared that the original SPT phase is at least partially protected by $G$, such that the left and right halves will be in distinct $G_L \times H$ phases, which is sufficient to imply an edge mode.

(weakly) explicitly broken—seems to be an under-appreciated and under-explored topic (however see the recent work Ref. [78] on strange correlators for higher SPTs). Indeed, it is a subtle one: which properties of the SPT phase do we expect to remain robust, if any?

One key property which certainly *does break down* upon explicitly breaking the protecting higher SPT is that it now becomes possible to adiabatically connect the SPT phase to a product state, at least for *periodic* boundary conditions. This is easiest to see for cohomology SPT phases, which admit an explicit circuit preparing them from a product state [71]. Since such a circuit is made up of local unitary gates, one can always turn it into a continuously evolving unitary which is connected to the identity[46].

Nevertheless, we argue that three non-trivial SPT properties are parametrically robust upon explicitly breaking higher symmetries (be they discrete or continuous): edge modes, entanglement modes, and the criticality between the SPT phase and the trivial phase. As we will note below, certain arguments in this section are not rigorous. Rather they should be treated as physical plausibility arguments, which we hope will inspire future works—both in the direction of trying to make these arguments more rigorous, as well as numerical studies which can test our stability arguments.

### 5.2.1 Edge modes

Consider an SPT phase protected (at least in part) by a higher symmetry. Its anomalous boundary can be in three possible states: gapless, spontaneous symmetry-breaking, or topologically ordered (more precisely, a deconfined gauge theory, allowing for the gapless case). Note that the later case is robust to any local perturbation, hence explicitly breaking the higher symmetry would still require a boundary phase transition to remove the edge mode (this part of the argument would apply to *any* SPT phase, not just higher SPTs). Let us thus discuss the two remaining cases: a gapless or symmetry-breaking edge.

If the SPT phase starts out with a symmetry-breaking edge, we need to distinguish whether it spontaneously breaks a higher symmetry or an additional global symmetry. In the former case, the boundary is topologically ordered (or more precisely, a robust deconfined phase, which might be gapless in the continuous symmetry case). If the boundary instead spontaneously breaks a global symmetry, then we are in the situation which was discussed in Sec. 4.4, such as for the 2D cluster state on the Lieb lattice, where the unbroken higher symmetry is essentially the order parameter of the spontaneous symmetry breaking at the edge, which is robust until we drive a boundary phase transition.

Finally, if the edge is gapless, we would argue that explicitly breaking the higher symmetry cannot trivialize the edge. This essentially hinges on the fact that in the IR, local relevant operators cannot be charged under a higher symmetry, since by definition the charged operators of a higher symmetry are not pointlike (see the review article Ref. [20] for a similar discussion in the context of bulk physics). Thus, any local operator which is able to gap out the gapless edge which is allowed when explicitly breaking the higher symmetry should *also* be allowed when we preserve the higher symmetry, but in that case the edge is anomalous, such that we can only gap out the edge theory into one of the above non-trivial gapped phases (i.e., symmetry-breaking or topologically ordered), for which we have already argued that one requires a boundary phase transition to destabilize them (even upon explicitly violating the higher symmetry). Admittedly, this argument would benefit from being made more rigorous, which might require new developments in the field of higher symmetries, which we leave to future work.

Hence, in all cases, we find that the SPT edge mode is parametrically robust upon

---

[46]More precisely, let $U$ be such a local unitary gate. We can write $U = e^{iH}$ for a local Hermitian operator $H$ (e.g., by diagonalizing $U$). We can thus define $U(\lambda) = e^{i\lambda H}$, where we can vary $\lambda \in [0, 1]$.

explicitly breaking the protecting higher symmetry. The edge cannot be immediately trivialized. Instead, one needs to drive a boundary phase transition, akin to what we saw in Fig. 1(a).

### 5.2.2  Entanglement spectrum

Another fascinating property of SPT phases arises in the bulk rather than the edge: upon bipartitioning the system in two halves, the entanglement spectrum is also constrained by an effective anomalous action. For instance, for the 1+1D SPT phase protected by global $\mathbb{Z}_2 \times \mathbb{Z}_2$ symmetry, one finds that the entanglement spectrum is twofold degenerate, due to a projective symmetry action [13]. Similarly, for the 2+1D cluster state on the Lieb lattice, the entanglement spectrum will be twofold degenerate—just like the symmetry-breaking edge theory we studied above. This is no accident: since the ground-breaking conjecture by Li and Haldane [79], it has been appreciated that the entanglement spectrum is dictated by an effective $(d-1)$-dimensional entanglement Hamiltonian, whose anomaly structure is exactly the same as the physical boundary Hamiltonian [80–83]. We note that for SPT phases, the entanglement Hamiltonian is indeed local, consistent with the fact that the wavefunction is obtained from a finite-depth circuit, and its locality has been explicitly confirmed (up to exponentially small tails), e.g., see Ref. [80].

Due to this intimate connection, the above stability arguments for the edge theory directly carry over to the entanglement spectrum. Note that this only implies non-trivial constraints on the *low-energy* part of the entanglement Hamiltonian. E.g., when we have the exact global and higher symmetries for the cluster state on the Lieb lattice, then the entanglement spectrum will be twofold degenerate for *all* levels, since there is an effective anomalous action on this entire (lower-dimensional or virtual) Hilbert space. In contrast, when we explicitly break the higher symmetry, our analysis in the previous subsection shows that the *lowest* levels of the entanglement spectrum remain twofold degenerate, and removing these requires driving the entanglement Hamiltonian to a phase transition.

### 5.2.3  SPT transition

Finally, let us consider SPT transitions. For instance, in Sec. 4.6 we saw that the cluster state on the Lieb lattice is separated from the trivial phase by a 2+1d (Ising × Ising)/$\mathbb{Z}_2$ criticality (see Fig. 7(a)). Moreover, as mentioned there, generic perturbations (preserving the protecting symmetry) would nudge this to $O(2)/\mathbb{Z}_2$ criticality. Now, what will happen if we explicitly break the higher symmetry? For global symmetries, there will generically be new relevant local operators we can add to theory, which would perturb it to a new fixed point (e.g., a trivial gapped phase). However, by the same plausibility argument as we discussed in Sec. 5.2.1, explicitly breaking a higher symmetry should not allow for new local operators. Hence, this implies we *cannot* immediately gap out the SPT transition. The only ways out are either eventually driving a first order transition, or we have to have another high-energy mode coming down in energy and tuning us through a multicritical point. This would be very interesting to study in future work.

## 6  Outlook

We have argued that the Higgs phase of a gauge theory can be best described as a non-trivial SPT phase protected by a higher magnetic symmetry as well as a physical global matter symmetry—where the instantiation of the latter depends on the particular context of interest, such as the physical symmetry imposed by an SIS junction, or the global

electric flux through the boundary of a system, or the relative charge between two Higgs fields. In the present work we studied in detail the case for discrete gauge symmetry, where we tested and confirmed the 'Higgs=SPT' proposal both using numerical lattice calculations as well as universal anomaly-type arguments. However, we stress that the general intuition (such as displayed in Fig. 2) is more generally applicable, and indeed, in Part II we explore the Higgs=SPT phenomenon for continuous gauge groups [27].

Our work suggests a variety of interesting questions to explore. For instance, in Fig. 1(b) we saw that the numerically-determined boundary phase transition of the Fradkin-Shenker model seems to roughly terminate at the bulk multicritical point—the latter was originally studied in Ref. [45, 46] and has proven to be an interesting topic in its own right, see e.g. the recent Ref. [84]. Our present numerics is not detailed enough to precisely resolve this feature, and it would thus be very interesting for future numerical studies to determine whether, indeed, the two match up. And if so, is there a simple reason for this?

A promising generalization would be to study the $U(1)$ lattice gauge theory [7]. Typically these do not have a limit where one has exact magnetic symmetry at the lattice scale (although see Ref. [85]), but we would still expect that deep enough in the Higgs phase, we should have boundary modes. More concretely, a numerical study could consider a 3+1D geometry with rough boundaries and investigate whether there is a 2+1D boundary superfluid. Indeed, the boundary Wilson operators (as in Fig. 2(c)) are local, gauge-invariant and charged, and could thus serve as a useful order parameter. Alternatively, one could further explore finite but non-Abelian gauge groups [86, 87].

It would also be interesting to investigate fermionic symmetry-protected topological phases due to Higgsing of gauge theories with fermionic matter that carries a gauge charge. An example of such a phase in one dimension realized in a $\mathbb{Z}_2$-gauged Kitaev chain has been already studied in [33]. Two-dimensional generalization of such a model or a $U(1)$-symmetric model studied in [88] might provide more robust examples of fermionic SPT protected by the combination of higher-form magnetic symmetry and global fermion parity. A closely related concept is the SPT cocycle related to the higher-dimensional fermionization/bosonization map [89–98].

One intriguing possibility opened up by our work is the stability of SPT phases protected (in part) by higher symmetries. In Sec. 5 we argued that various properties of interest would be parametrically robust to explicit breaking of the protecting higher symmetries. It would be worthwhile to test numerically these ideas in future works. A natural starting point is the $O(2)/\mathbb{Z}_2$ criticality arising for the Lieb lattice cluster state (see Fig. 7 and the discussion in Sec. 4.6.1). One could numerically study the fate of this criticality upon explicitly breaking the loop-like symmetry of this lattice model; our arguments suggest it should remain a codimension-2 critical point for a range of perturbation strengths, and eventually it should make way to a new multicritical point.

One possible connection to explore further is with recent work on using measurement to prepare long-range entangled states in finite time [48, 99–111]. In particular, Ref. [103] introduced the idea that measuring certain charges of an SPT phases can lead to long-range entanglement for the remaining qubits. One of the simplest examples is measuring the vertices of the cluster state on the Lieb lattice, which produces toric code order on the links [99]. In the light of Higs=SPT, we can interpret this mechanism as starting in a trivial phase were charges have condensed, and measurement subsequently pins the location of the gauge charges, thereby again deconfining the magnetic particles, leading to the topological (deconfined) phase. It would be interesting to explore more generally how measuring Higgs phases can lead to deconfinement (in fact Ref. [103] points out that measuring the Gauss law operator can be used to gauge any Abelian symmetry), and formulate it in a high-energy field-theoretic language. A related question is: we have

showed that various SPT properties remain robust upon explicitly breaking the higher symmetry; does this imply that measuring the gauge charges in an open region akin to the blue shaded region in Fig. 1(b) will produce robust long-range entanglement in the post-measurement state? This indeed seems to bear out, as will be explored in an upcoming work motivated by an entirely different angle, namely that of sweep dynamics in Rydberg atom arrays [112].

# Acknowledgements

We thank Ehud Altman, Erez Berg, Daniel Harlow, Eduardo Fradkin, Ho Tat Lam, Leon Liu, Rahul Sahay, Nat Tantivasadakarn, Juven Wang, and Cenke Xu for insightful discussions. In particular, we thank Tibor Rakovszky for extensive discussions, and for collaboration on the closely related Part II [27]. Some of the discussion of an emergent gauge theory (as in Sec. 3) is heavily inspired by lecture notes by John McGreevy [113]. RV is supported by the Harvard Quantum Initiative Postdoctoral Fellowship in Science and Engineering. RV and AV are supported by the Simons Collaboration on Ultra-Quantum Matter, which is a grant from the Simons Foundation (651440, A.V.). RT is supported in part by the National Science Foundation under Grant No. NSF PHY-1748958. UB is funded by the Deutsche Forschungsgemeinschaft (DFG, German Research Foundation) under Emmy Noether Programme grant no. MO 3013/1-1 and under Germany's Excellence Strategy - EXC-2111 - 390814868. SM is supported by Vetenskapsrådet (grant number 2021-03685).

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

# A    Numerical details for the boundary transition

In Fig. 11 we present the numerical details for the observables discussed in the main text in Sec. 4.4.3. Most notably, we can identify the region of the Higgs phase where the boundary spontaneously breaks the $P$ symmetry by measuring a local order parameter along the rough boundary. We performed iDMRG simulations for strips over varying width, and we find that already for the results reported in Fig. 11 the dependence on the width is negligible, such that we obtain a good approximation for the boundary transition of the 2D system.

# B    Fradkin-Shenker model with a smooth boundary

In Sec. 4.4 we considered the 2+1D Fradkin-Shenker model with rough boundaries, which led to protected SPT edge modes in the Higgs phase. Here we instead consider the smooth

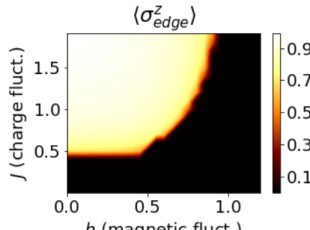 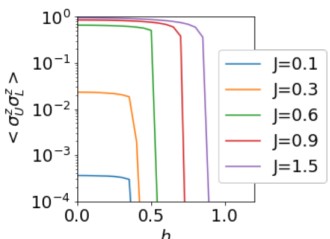

Figure 11: Numerical iDMRG results for a ladder with $L_y = 3$ plaquettes and rough edges. Left: the edge order parameter $\langle \sigma_{edge}^z \rangle$ has a finite value in the Higgs phase, indicating the robustness of edge modes even in the presence of the perturbation $h$, which explicitly breaks the magnetic symmetry $W$. Right: the transition can also be detected by the correlator $\langle \sigma_U^z \sigma_L^z \rangle$ between the upper and lower edge of the ladder. This is non-zero even in the deconfined phase (although exponentially suppressed in $L_y$), and therefore it singles out the confined phase.

boundary. As explained in Sec. 4.4.5, edge modes will still exist if we respect the magnetic symmetry. To derive such a natural boundary Hamiltonian, we use the approach advocated in Sec. 4.6.3, where we interpret a boundary as a spatial interface to a distinct SPT phase. For concreteness, we take the emergent gauge theory introduced in Sec. 4.6.1

$$H = -\sum_v X_v - \sum_p B_p - J \sum_{\langle v,v' \rangle} Z_v \sigma_{v,v'}^z Z_{v'} - K \sum_v G_v - \frac{1}{K} \sum_l \sigma_l^z - h \sum_l \sigma_l^x \quad (30)$$

and drive the upper half of our system into the trivial product state with each link (vertex) variable $\sigma_l^z = 1$ $(X_v = 1)$[47] :

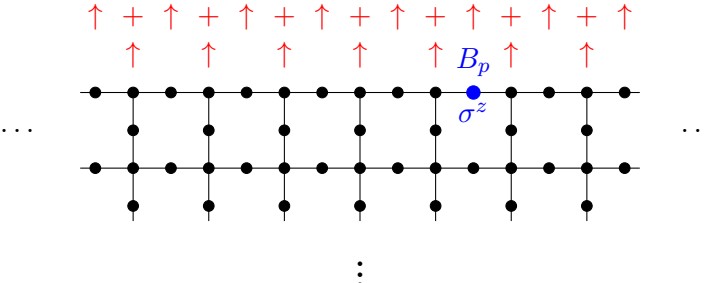

We see that the four-body plaquette term $B_p$ is effectively reduced to a single-site term at the 'boundary' (i.e., at the spatial interface). Secondly, we lose the energetic term enforcing the gauge constraint on the outer (matter) sites, since it does not commute with the link variables where we drove $K \to 0$. However, we of course still have the (emergent) gauge constraint on all the other vertices; using these to eliminate the matter fields, we obtain an effective link model with additional matter sites on the outer edge:

---

[47]The former is achieved by taking the limit $K \to 0$ in Eq. (30)

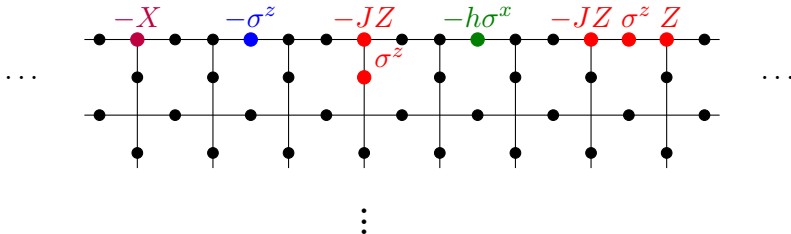

We schematically show the boundary terms that appear in this effective model. As for the effective $P$ action: it acts with $\prod X$ on the matter sites and as $\prod \sigma^x$ on the *second* row:

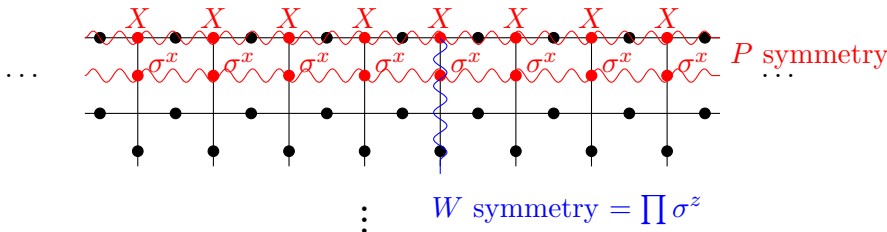

We see that $W$ and $P$ still anticommute, giving us the protected degeneracy. (Of course, $h_x \neq 0$ will break $W$, but we know the edge will be in the $P$-breaking state, which is locally stable.)

To understand how stable this is, it is interesting to observe that for large $h_x^2 + J^2$ (i.e., at large radii in the Fradkin-Shenker phase diagram) the bulk becomes a product state. More precisely, the bulk Hamiltonian is simply $H_{\text{bulk}} \propto -\sum_l \boldsymbol{n} \cdot \boldsymbol{\sigma}$ where $\boldsymbol{n} = (\sin \varphi, 0, \cos \varphi)$ where $\varphi = \arctan(h_x/J)$. Hence, our system effectively only consists of two remaining rows! Furthermore, for any Hamiltonian term, we can project them down by using $\langle \sigma^x \rangle_{\text{bulk}} = \sin \varphi$ and $\langle \sigma^z \rangle_{\text{bulk}} = \cos \varphi$. For example, the star and plaquette terms just below the edge reduce in this limit to:

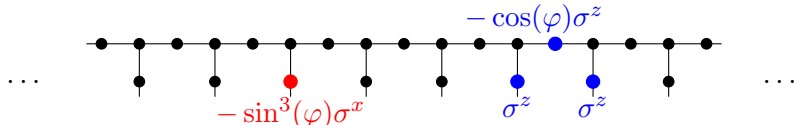

It is easier to summarize the effective boundary Hamiltonian if we first introduce the following notation for our three-site sublattice:

$$
\begin{array}{c} \text{B} \;\; \text{C} \\ \bullet\!-\!\!\bullet \\ \bullet \\ \text{A} \end{array}
$$

Using the $X, Y, Z$ as a notation for the Pauli matrices, we have the following boundary Hamiltonian, which is *exact* for large $h_x^2 + J^2$:

$$
H_{\text{smooth bdy}} = -\sum_n \Big( X_{B,n} + h_B Z_{C,n} + J Z_{B,n} Z_{C,n} Z_{B,n+1} + J Z_{A,n} Z_{B,n} + h_x X_{C,n} \tag{31}
$$

$$
+ \big( h_x + \sin^3 \varphi \big) X_{A,n} + h_B \, \cos \varphi \, Z_{A,n} Z_{C,n} Z_{A,n} \Big).
$$

To get some understanding from this daunting-looking spin chain, note that if one only includes the dominant energy scales (i.e., $h_x$ and $J$ since—remember—we are working at large radii in the phase diagram), we actually have a local conserved quantity: $X_{C,n-1} X_{A,n} X_{B,n} X_{C,n}$, which can be interpreted as the boundary Gauss operator. Let us

denote this conserved quantity by $\tilde{X}_n$. We will derive the effective low-energy Hamiltonian for this Hilbert space.

When $J \gg h$ is the dominant energy scale, then one obtains an effective Hamiltonian at fourth order in perturbation theory:

$$\lim_{1,h_B \ll h \ll J} H_{\text{bdy}} \propto -\sum_n \left( h_B \tilde{Z}_{n-1} \tilde{Z}_n + \left( \frac{h}{J} \right)^3 \tilde{X}_n \right). \tag{32}$$

This clearly has the critical point at $h/J = \sqrt[3]{h_B}$, so this perturbative expansion is only valid for small $h_B \ll 1$.

When $h \gg J$ is the dominant energy scale, we see that essentially $X_A$ and $X_C$ get pinned to $+1$. Hence, $\tilde{X}_n = X_{B,n}$. We now derive the following effective Hamiltonian at second order in perturbation theory:

$$\lim_{1,h_B \ll J \ll h} H_{\text{bdy}} \propto -\sum_n \left( \frac{J h_B}{h} \tilde{Z}_{n-1} \tilde{Z}_n + \tilde{X}_n \right). \tag{33}$$

Hence, now the critical point is at $h/J = h_B$, which is only consistent with this expansion in $J/h$ if $h_B \gg 1$ is large.

We have numerically confirmed these perturbation theory results by simulating the full model in Eq. (31) using DMRG. However, neither of these perturbative results tells us what happens when $h_B = 1$. Fortunately, DMRG gives us a clear answer: the boundary criticality seems to stable up to $h_x \approx 1.5J$. Remarkably, this means it is stable beyond the duality line of the Fradkin-Shenker model.

## C   Bulk-defect correspondence

Consider a codimension $k$ defect along a subspace $X \cong \mathbb{R}^{d-k+1}$ in $d+1$-dimensional spacetime $\mathbb{R}^{d+1}$. For $k > 1$, $\mathbb{R}^{d+1} - X$ is connected, so there is no local condition on the defect which can prevent charge from going from one side to the other—it can just go around. However, $\tilde{H}_{k-1}(\mathbb{R}^{d+1} - X, \mathbb{Z}) = \mathbb{Z}$, so we can impose a generalized no-tunnelling condition which preserves the charge of extended objects with $k$-dimensional worldvolumes which may become trapped "around" the defect.

If these extended objects carry charge under a $k-1$-form symmetry, it means the intersection number of their worldvolumes with any closed codimension $k$ surface in spacetime (the "flux" through the surface) is conserved. The generalized no-tunneling condition ensures that the flux through a surface with boundary on the defect is also conserved. This defines an ordinary $G$ symmetry on the defect. See Fig. 12.

The anomaly on the defect may be computed by compactification. For simplicity let us consider a product symmetry $G \times E[k-1]$ and SPT class $\omega_{d+1} \in H^{d+1}(G \times E[k-1], U(1))$. We use group cohomology notation but the result is the same for other cohomology theories like cobordism. We define a codimension $k$ defect by forbidding tunneling of $E[k-1]$ charges across it. The symmetry on the defect is $G \times E[k-1] \times E$, where the extra factor of $E$ comes from the no-tunneling condition.

We wish to describe the induced anomaly on the defect as an SPT class

$$\omega_{d-k+2} \in H^{d-k+1}(G \times E[k-1] \times E, U(1)). \tag{34}$$

Suppose $Y^{d-k+2}$ is a $d-k+2$-dimensional spacetime equipped with a $G \times E[k-1] \times E$ gauge field $(A, B_k, C_1)$. We use this to define a $G \times E[k-1]$ gauge field $(A', B'_k)$ on

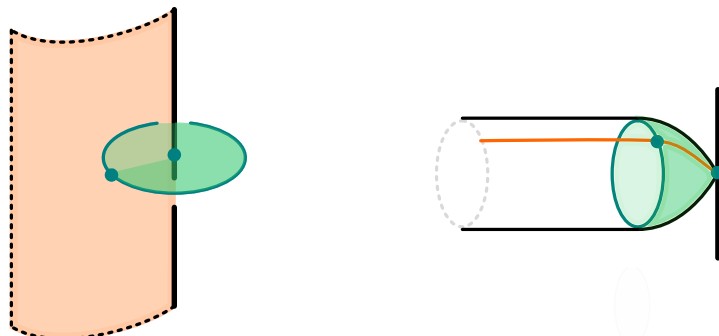

Figure 12: The generalized no-tunneling condition ensures the conservation of extended charged objects wrapping the defect (solid black line), as measured by intersection with the higher form symmetry operator on a sheet (orange) ending on the defect. This forbids tunneling operators (green disc) which would change this intersection number. Compactifying the normal coordinates we realize the defect as a boundary condition of a $d - k + 1$ (space) dimensional SPT where the higher form symmetry operator acts as a 0-form symmetry, which allows us to determine the anomaly on the defect.

$Y^{d-k+2} \times S^{k-1}$. Let $\pi : Y^{d-k+2} \times S^{k-1} \to Y^{d-k+2}$ be the projection map and $v_{k-1} \in H^{k-1}(Y^{d-k+2} \times S^{k-1}, \mathbb{Z})$ be the volume form of $S^{k-1}$. Then we define

$$A' = \pi^* A$$
$$B'_k = \pi^* B_k + C_1 \cup v_{k-1}$$
$$\int_{Y^{d-k+2}} \omega_{d-k+2}(A, B_k, C_1) := \int_{Y^{d-k+2} \times S^{k-1}} \omega_{d+1}(A', B'_k). \tag{35}$$

This calculation reproduces only part of the SIS junction anomaly for $k = 1$. The integration on $S^0$ computes the difference in the SPT class from either side of the junction. That is, although we have independently $E_L \times E_R$, this method sees only the skew diagonal subgroup $E \simeq \{(e, e^{-1}) \in E_L \times E_R | e \in E\}$.

Anomalies of a simple product form $\omega_{d+1} = \omega_{d-k+2} \cup B_k$ can always be detected since $\omega_{d-k+2}$ is the anomaly on the defect. An example of an anomaly which cannot be detected this way is the order 2 element of $H^4(\mathbb{Z}_2[1], U(1)) = \mathbb{Z}_4$, given by $\omega_4 = \frac{1}{2} B_2^2$. Abstractly one can see this because the reduction to the defect defines a homomorphism $H^4(\mathbb{Z}_2[1], U(1)) \to H^2(\mathbb{Z}_2[1], U(1)) = \mathbb{Z}_2$, which must have a kernel $\mathbb{Z}_2$ generated by this element. The problem is that we are evaluating $\frac{1}{2} B_2^2$ on 4-manifolds of the form $\Sigma \times S^2$, with $B_2 = x_\Sigma + y_{S^2}$, where $x_\Sigma \in H^2(\Sigma, \mathbb{Z}_2)$ and $y_{S^2} \in H^2(S^2, \mathbb{Z}_2) = \mathbb{Z}_2$ is the generator. We have

$$B_2^2 = (x_\Sigma + y_{S^2})^2 = x_\Sigma^2 + 2 x_\Sigma y_{S^2} + y_{S^2}^2 = 2 x_\Sigma y_{S^2} = 0 \pmod 2, \tag{36}$$

so the defect cannot see this particular anomaly.