# Peer review of "Higgs Condensates are Symmetry-Protected Topological Phases: I. Discrete Symmetries"

_SciPost Physics_

## Round 1 · Referee Report · Anonymous (Referee 1) · 2024-4-30

Strengths

  1. Topic wise, this paper brings points out an important nature of the Higgs phase as a SPT phase (partially) protected by higher-form symmetries. This is an intriguing point of view that has not been explicitly pointed out before and leads to interesting theoretical consequences.

  2. The tone of writing in the paper is rather brisk and refreshing.

  3. The paper is organized in a quite accessible fashion for general audience by starting from a simple example that demonstrates most of the important results.

Weaknesses

  1. The paper struggles with how detailed the analysis should be for each cases. The models discussed in Sections 3 and 4 are rather standard textbook models with only minor tweaks, and it seems that the authors have spent too much time talking about different physical aspects of the same phenomena.

However, Section 5 does provide a lot of interesting new insights in our understanding of SPT states, but the amount of words spent over there is quite small, comparing to the length of the paper. There are important open questions in section 5 that have not yet been thoroughly worked out.

It seems that the authors want to write an eye-catching paper with a strong punchline, but in doing so, a lot of important theoretical and realistic questions are glossed over, while the readers find themselves spend most of their time wandering between different sides of a same coin.

Report

I believe this paper generally meets the acceptance criteria of SciPost Physics, after these following concerns being addressed.

  1. Right before Sec. 2.1 " In this sense, the Gauss law is an SPT stabilizer."

Readers who are not familiar with the stabilizer formalism of quantum error correction code or the fixed point Hamiltonian of an SPT state will struggle to see this point.

  1. Sec. 3.2.1 "From Bob’s point of view, his universe is indistinguishable from a conventional (lattice) gauge theory..."

What is different from a conventional lattice Z_2 gauge theory here? The Hamiltonian in (5) is simply how we gauge the global Z_2 symmetry, denoted by P in the paper's notation. In the usual gauging procedure one can also fix the global gauge sector in 1d by eigenvalue of W.

  1. Sec. 3.2.4 " Such a string order does not imply any bulk degeneracy, and indeed Eq. (12) is still constrained by the global condition ..."

The resolution of the misinterpretation by Bob is not really clear from this sentence. The reader might be mislead into thinking that no bulk degeneracy in the gauge theory is the same as (12) having a unique ground state on a finite size system.

  1. Sec. 3.2.6 "This particular SPT transition has a central charge c = 1"

Citation is probably needed on this one.

"One can say that the two Higgs phases differ by an SPT phase."

It does not seem clear that large J' is a non-trivial SPT phase from the Hamiltonian in (16). And what SPT phase are they differ each other from? From (16) the difference betweeb large J and J' is an extra global controlled gate between each neighboring pair of \sigma^x. But this SPT entangler is trivial since there is no SPT state protected in 1d by a single Z2 symmetry.

  1. Sec. 4.3 " In particular, we define ..., where CZ is the “Controlled-Z” gate"

Isn't U just a product of CNOT cates? Is there any specific reason why CZ is preferred over CNOT in this case?

  1. Sec. 4.4.1 "In contrast, in the Higgs phase, a W line can terminate in the bulk (Eq. (21), or more precisely see footnote 27), since we have a charge condensate, leading necessarily to a twofold degeneracy. Moreover, since the bulk is short-range entangled, this degeneracy is located entirely on the edge (in contrast to the topological degeneracy of the deconfined phase)."

The authors should consider elaborating on the penetration depth of edge mode/degeneracy into the bulk.

  1. Sec. 4.6.1 "This splits apart into four distinct Ising critical lines; more precisely, two of these are the conventional 2+1D Ising criticality, whereas the other two are the gauged version. Moreover, the ones neighboring the Higgs phase have non-trivial symmetry-enrichment due to P and W symmetry, which will protect edge modes even at criticality"

The meaning of Ising* and Ising/Z2 in Fig 7(a) is not defined in the text. These identifications needs more elaborations, they are not obvious from the text.

  1. Sec. 4.8 "We illustrate this with an example. Let X and Z denote the two clock operators for Zn (for n = 2 these reduce to the Pauli operators) ..."

The way this section is organized is a bit confusing. A more logical way of presentation should be that we first write down the anomalous symmetry of this Z_n\times Z_n[d-1] mixed anomaly, and then writing down this Hamiltonian to justify that this symmetry-allowed Hamiltonian leads us to an SSB state. And then talk about the robustness of the constraining power of the mixed anomaly.

Also, it is incorrect to say Z commuts with \prod_i X_i for n\neq 2. It is better to just write down the algebra between these two.

" Amat and Bmag can be interpreted as gauge fields for the Zn matter and Zn[d − 1] magnetic symmetries, respectively, and 2πΩ as a topological term in the effective action of the Higgs-SPT bulk. "

It would be better if one just write down the effective action instead of glossing it over using words: the partition function with the presence of background gauge field is (-1)^{i 2 \pi \Omega}.

  1. "For definiteness, let us say the total symmetry group protecting the SPT phase is G×H. The simplest and most direct way of arguing the above bulk-defect correspondence is by reducing it to the usual case of an SPT edge mode."

In this paragraph it is better to state in the beginning that the SPT phase is classified with a non-trivial cocycle between some elements in G and H, so that the anomaly on the (d-1)D domain wall originates from the mixed anomaly between G and H.

Although I believe that the picture in this section is physically correct, it is a rather big jump from Higgs-SPT phase to a generic 0-form SPT state. Higgs-SPT state is a small class of SPT states protected by a 0-form symmetry and a higher-form symmetry whose group is Pontryagin-dual to the 0-form symmetry group. There is only a 2-cocycle between these to symmetries in any spacial dimension, which makes this no-0-form-charge-tunneling defect straightforward to discuss. For generic SPT states with higher cocycles, I do not see an easy way to visualize this. The authors should consider adding a concrete example (maybe after Appendix C) of a 0-form symmetry protected SPT state in 2+1 or 3+1D, and the form of the anomalous symmetry action explicitly.

Requested changes

See report

Recommendation

Ask for minor revision

---

## Round 1 · Referee Report · Anonymous (Referee 2) · 2024-5-13

Strengths

  • Deep thinking of commonly overlooked physics in familiar models. The Higgs=SPT viewpoint can predict interesting new physics such as the boundary phase transition between the Higgs and confined phases in the Fradkin-Shenker model.
  • Conveys the messages clearly through simple examples.

Weaknesses

  • Both sections I, II are introductions. They are too long and can be distractive. While the authors made an effort to explain things as clear as possible without invoking equations, the long paragraphs actually hinder the understanding. I found section III much more helpful than sections I, II. It is worth considering merging sections I, II or sections II, III.
  • In general this paper is too long and contains too many subsections. While the viewpoint Higgs=SPT is important and deep, it is a simple point to make and these long discussions seem unnecessary.

Report

This work proposes to view the Higgs phase in a gauge theory is an SPT protected by a higher form magnetic symmetry and a matter symmetry, and discusses various consequences and predictions. The results are very interesting and important.

A few comments: 1. Bottom of page 4, about Goldstone mode. Consider a FM Heisenberg model living in high spatial dimensions, and add a random field which breaks the SU(2) spin rotation symmetry down to a weak symmetry (the random field averages to zero). While the symmetry is explicitly broken, the added term can be irrelevant in the IR and still leads to the conventional Goldstone mode. Therefore, the continuous symmetry is emergent in the same sense how higher-form symmetries can be emergent in the IR while explicitly broken in the UV.

  1. The authors repeatedly emphasize how the SPT properties are robust against explicitly breaking the magnetic symmetries, but this is not something new - it is well known how higher form symmetries can be explicitly broken in the UV but emergent in the IR (such as in topological phases). It would be helpful to point out this emergence point of view that people are familiar with.

  2. Typo: last paragraph above acknowledgements, "were charges have condensed" -> "where charges have condensed".

Requested changes

See the "weaknesses" section and the report above.

Recommendation

Publish (surpasses expectations and criteria for this Journal; among top 10%)

---

## Editorial Decision

awaiting_resubmission